# Detecting anthropogenically induced changes in extreme and seasonal evapotranspiration observations

**Marius Egli** [1] ✉, **Sebastian Sippel** [2], **Reto Knutti** [1] **& Vincent Humphrey** [3]

Increasing temperature and radiation drive an increase in evaporative demand. However, it is still uncertain whether the increase in demand has led to an increase in evapotranspiration (ET) in observational products, as this increase is at odds with a limited water supply over land. Here, we examine changes in high ET extremes and seasonal mean ET using climate models as well as observational data. High ET extremes are driven by periods with high incoming surface radiation and temperatures. In line with physical understanding, these events are intensified by anthropogenic climate change. We detect robust changes in extreme and seasonal ET in two observational data sets. Regionally, seasonal mean ET shows mixed increases and decreases from 1980 to 2023, while extreme ET universally increases or shows no significant change. Although the drivers for these changes can vary regionally, we expect that regions with strong extreme ET trends are at increased risk of flash droughts.

The alteration of the hydrological cycle is a fundamental consequence of climate change. A central hydrological variable affected by these changes is evapotranspiration (ET). Linking the water, energy, and carbon cycle, ET is a key to understanding the effects of climate change. From physical first principles, we know that the water holding capacity of air increases alongside temperature[1,2]. We therefore expect an increase in potential ET, which quantifies the amount of water that could evaporate in the absence of water limitation. Increasing ET could drive an increase in drought risk, which warrants further investigation[3]. However, studies in climate models have shown that ET and potential ET diverge in future scenarios[4], which makes it crucial to study ET itself[5].

Another process driving the drought risk is the increase in the variability of precipitation[6,7] and evaporation[8,9]. The increase in precipitation variability is the result of extreme precipitation increasing faster than mean precipitation, since extremes react more directly to the increase in atmospheric water holding capacity[10,11]. Whether this increase in variability also applies to ET is less clear, but recent evidence in central Europe suggests this is the case[12]. However, research on changes in seasonal or yearly mean ET does not find universal global

increases. Rather, they find complex patterns of increases and decreases[13], which is a testament to the complex processes that affect ET in a changing climate. A central limitation for ET is the availability of water and energy[14], which varies depending on the regional hydrological conditions. Regions where water is scarce are not expected to see increases in ET, as there is simply not enough water to sustain an increase. Regions receiving more precipitation can be limited by the energy available for ET. For such regions, an increase in potential ET is expected to result in higher actual ET.

Historical changes in ET have been studied with a focus on annual mean ET, where many regions show increases[15,16]. However, there have been indications for a leveling of this trend due to water limitation[17,18]. An increase in observational annual mean ET has recently been detected and attributed to anthropogenic activity[19].

Although changes in ET should be easier to detect than precipitation and runoff due to the higher signal-to-noise ratio[20], there remains substantial uncertainty for multiple reasons. Firstly, measuring ET is complicated, and more widespread data have only been collected since the late twentieth century. Observational products with global coverage can only be provided with the help of physical or

[1]Institute for Atmospheric and Climate Science, ETH Zurich, Zürich, Switzerland. [2]Institute for Meteorology, Leipzig University, Leipzig, Germany. [3]Federal Office of Meteorology and Climate MeteoSwiss, Zurich, Switzerland. ✉e-mail: marius.egli@env.ethz.ch

statistical models. These models rely on a set of assumptions and contain known biases[21]. Furthermore, these models often require additional data, such as Supplementary ET data, adding additional uncertainty[22]. These aspects limit the amount of reliable observational data available for long-term analysis, making the extraction of an anthropogenic signal increasingly challenging compared to variables such as temperature. Secondly, ET is difficult to accurately model in climate simulations because it is highly sensitive to the representation of soil and plant dynamics in these models[23]. Climate models often diverge in their future projections of ET and soil moisture[24]. Furthermore, ET is affected by the stability of the atmosphere, which is only approximately represented in climate models. These challenges make it difficult to assess the impact of climate change on ET. Third, ET is a quantity that integrates the effect of many different processes in the climate system. It is sensitive to energy input at the surface, water availability, stomatal conductance, and atmospheric conductance. All of these are subject to changes due to anthropogenic influence, and the change in ET is the sum of all of these changes. Finally, hydrological variables are subject to substantial internal variability[25]. Thus, it is challenging to detect a trend that is robustly outside the range of what can be induced solely by natural variability.

Although these challenges complicate the identification of robust trends in ET observations, we hypothesize that the constraint of water limitation is weakened when focusing on high ET extremes, reducing the impact of the confounding variable. In addition, recent studies have been able to use statistical learning approaches in combination with climate model data to robustly detect anthropogenically induced changes in the presence of large natural climate variability, which will be crucial in this context[26,27]. Thus, we test whether projected increases in potential ET manifest themselves in ET when only considering the seven consecutive days with maximum ET. Are there detectable increases in the available observational products for the magnitude of high ET extremes? Such an increase in ET would lead to a faster transition from normal to dry conditions, speeding up the development of droughts, where high ET can be a characteristic feature and a central driver[28,29]. This seems especially relevant to research on flash droughts, which has become increasingly relevant in the last decade. Based on recent findings on the change in the timing of extreme precipitation events[30], we hypothesize that the timing of extreme ET events could shift to the wet season, driven by constraints in water availability.

In this study, the notion of a high ET extreme is established, and examples of recent events over a region in Europe are provided. We investigate the evolution of intensity and timing of such events in CMIP6 climate models. We then apply methods from detection and attribution studies to find trends in extreme and seasonal ET on a global and hemispheric scale in observations, and test the robustness of these trends by comparing them to trends in CMIP6 model simulations. Finally, we identify regions that have experienced robust changes in extreme or seasonal ET.

## Results
### What is an extreme ET event?
To illustrate what an extreme ET event is, we first consider the seven days of every year when a region in Central Western Europe experienced the highest cumulative ET (ETx7d) in ERA5 Land (Fig. 1). We highlight the years with the strongest (2019) and second strongest (2022) ETx7d events. These seven days are commonly associated with little to no precipitation, as well as high incoming short-wave radiation. In addition, temperatures tend to be high and the relative humidity low, leading to a high vapor pressure deficit. In the mid latitudes, these conditions are often associated with high surface pressure, but this is not necessarily found in tropical regions such as eastern Brazil (Fig. S6). These events result in strong decreases in soil moisture.

The evolution of these variables points to a heat wave as the common driver, which matches our physical expectation of such an

event[31]. The two strongest events over Europe are characterized by very high short-wave radiation, low relative humidity, and high temperatures. We also find these features in other regions, like eastern Brazil (Fig. S6) and the central US (Fig. S5). In dryer regions like the central US, we also see a preconditioning on water availability. The two strongest events occurred at relatively high soil moisture leading into the event. In contrast, the year 2012, which was associated with a strong drought, had one of the smallest ETx7d values. In the European study region, the two strongest events occurred very recently, suggesting an increase in ETx7d. We therefore want to investigate whether we find this tendency towards increasing ETx7d with CMIP6 models.

### Event intensity in CMIP6
For this, we first consider the evolution of the intensity of an ETx7d event in climate models forced with the historical and then the SSP5-8.5 emission scenario in the European SREX regions (Fig. 2). In the Mediterranean, most climate models show a decrease of ETx7d intensity up to 1980, with the exception of CESM2, where the intensity increases up to around 2030. The period between 1950 to 1980 is associated with high aerosol emissions, which dims short-wave radiation and thus limits ET. The subsequent reduction in aerosol emissions and therefore increased short-wave radiation, most models show an increase in ETx7d. Event intensity in UKESM1-0-LL remains low and never recovers to levels of 1850. Although the models span a wide range of responses, the observations agree on an increase, after a local minimum around the year 2000. However, hydrological variability is particularly large in this region[32] and trends should be interpreted with caution. In Western Central Europe (WCE), the models behave similarly to those in the Mediterranean. The increase in ETx7d intensity after 1980 is more pronounced, but so is a decline starting in 2020. CESM2 again shows no local minimum, and ETx7d increases continuously until 2050. However, all models agree on an increase in ETx7d between 1980 and 2020. The decrease, which all models show, is likely tied to water limitations emerging towards the end of the century in a high-emission scenario[33]. The observations all agree on very strong increases starting in 1980, which again points to the importance of aerosol-induced changes. ETx7d in northern Europe decreases slightly during the period of high aerosol emissions in Europe, reaches a minimum in most models around 1980 and then increases above the levels early in the simulations. MPI-ESM1-2-HR and CESM2 show slight decreases after 2050 in ETx7d, while the other three models show steady increases. However, the evolution over time is very comparable between climate models. The trends in the observational products are not as strong as for WCE but seem to intensify towards the most recent years.

### Event timing in CMIP6
We would expect increasing water limitation to result in ETx7d events shifting towards earlier in the season, when there is more water available. However, this cannot be concluded from the data, as the timing of ETx7d events is subject to less clear and intense changes in general. In the Mediterranean, the climate models show a tendency towards events earlier in the season towards the end of the 21st century. The observations, but also the climate models, show substantial disagreement in the overall timing. Events in GLEAM and ERA5 Land tend to occur comparatively early in the season. The only climate model that shows similarly early extremes is MPI-ESM1-2-LR. Events in X-BASE occur later in the season, but still not as late as two of the climate models shown here. The reason for this large range in timings is likely caused by discrepancies in the strong seasonal cycle of ET. Since we are extracting the highest absolute values, they can realistically occur only during the peak of the seasonal cycle. Climate models have been shown to have discrepancies in the seasonal cycle of precipitation[34], which will have the biggest impact in a more water-limited Mediterranean. In WCE, there is broader agreement on the timing of ETx7d events. Most climate models show a tendency towards

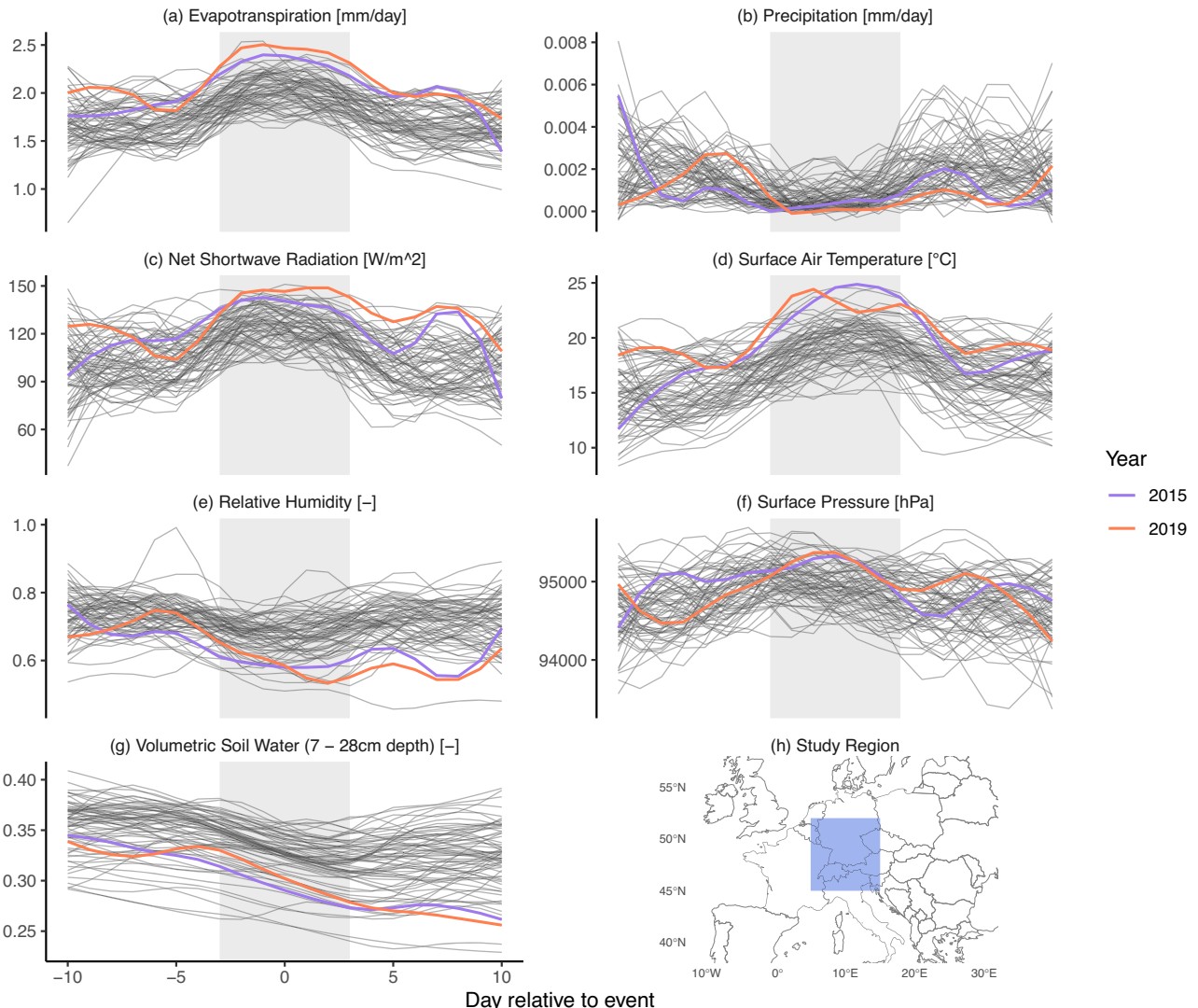

**Fig. 1 | Composite time series of various variables during an extreme ET event in ERA5 Land.** The x-axis indicates days relative to the event, with 0 being the center of the 7-day event. The gray box indicates the 7-day duration based on which the events were selected. The variables shown are **a** evapotranspiration, **b** precipitation, **c** net short wave radiation, **d** surface air temperature, **e** relative humidity, **f** surface pressure, and **g** volumetric soil water. Two lines are colored and indicate the strongest (2019) and second strongest (2015) events in the study region (**h**). The other lines indicate the evolution of all other events between 1950 and 2023. A loess smoother was applied to improve visual clarity.

earlier events towards the end of the 21st century, which again points towards water becoming an important limitation. GLEAM and ERA5 Land do not show great changes in the timing, in contrast to X-BASE, which sees a shift towards events earlier in the season. In Northern Europe, the climate models again show little change in the event timings. The observations lie at the lower end or even outside the range spanned by the climate models.

Overall, we see relatively broad agreement among climate models and observations on an increase in ETx7d intensity over the recent historical period, which we want to investigate on a larger spatial scale. We want to see whether a detectable shift in the distribution of ETx7d has been observed and whether such a shift is consistent with climate models.

### Detecting changes in extreme ET

To detect a change in the observational data sets, we employ a detection and attribution technique based on a regularized regression model. The regression model predicts the ensemble mean ETx7d of a climate model from the ETx7d maps of only a single ensemble

member. Such a regression setup results in a statistical model that is able to reduce the influence of internal variability but also accounts for some climate model disagreement, allowing for a more robust trend detection. In order to examine whether changes in the extremes are stronger than in the mean, we repeat the analysis for the mean ET of the Northern hemisphere of June, July and August (JJA) and the mean ET of the Southern hemisphere of December, January, and February (DJF). The coefficients of the regression models, shown in Fig. 3a, b, c, are mostly positive. They can be mapped out to their corresponding location, showing where the model is drawing information from. Grid cells that are assigned a large coefficient contribute more to the prediction of the forced response (FR). These grid cells are in regions where internal variability is comparatively low and where climate models agree most about their relationship with the ensemble mean. We do expect mostly positive coefficients, since the ensemble mean is positively correlated with the grid cells that underlie it. Using this regression model, we then predict the forced response for all the members of the climate models used and calculate the 1980–2023 trends forced with historical emissions. Additionally, we also predict

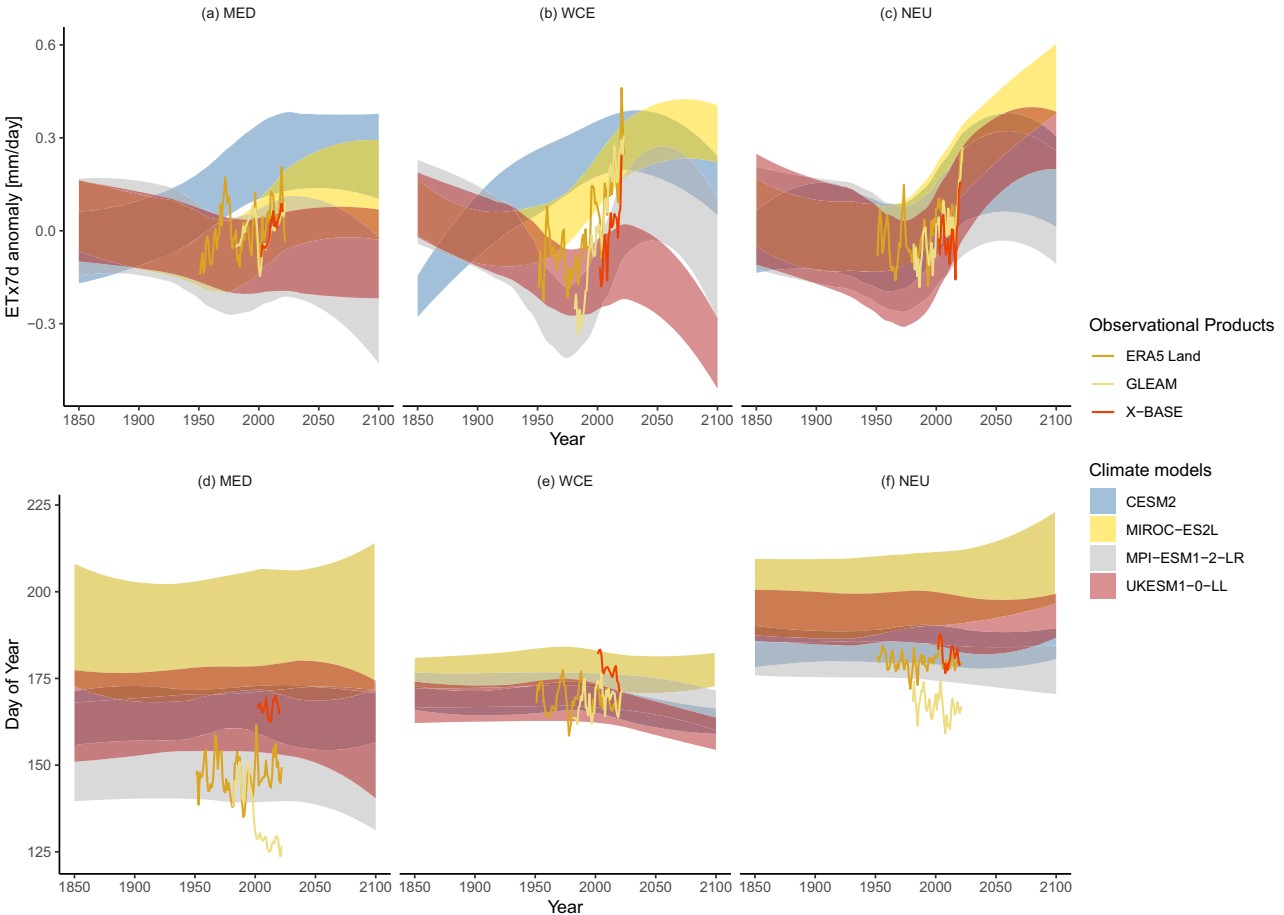

**Fig. 2 | Time series of climate model and observed ETx7d anomalies and timing.**
Anomalies are shown for the three European SREX regions Mediterranean Med (**a**),
Western Central Europe WCE (**b**), and Northern Europe NEU (**c**). The shading shows
a smoothed range of ± one standard deviation around the ensemble mean in a
selection of climate models forced with the historical and subsequent SSP5-
8.5 scenario. The lines represent the Ex7d in three observational products over
these SREX regions with a three-year running mean applied for visual clarity.
**d**, **e**, **f** show the timing as the day of the year when the maximum occurred in the
same climate models and observations. Day 125 corresponds to 5 May and day 225
to 3 August.

the forced response in climate model simulations, which exclude
anthropogenic activity (PiControl), to estimate the ranges of likely
unforced trends (Fig. 3d, e, f). The unforced trends serve as a null
distribution of trends that can emerge solely due to natural variability.
The historical trends serve as the alternative hypothesis, representing
the ranges of trends we can expect under anthropogenic activity. We
then applied the regression model to the two observational data sets,
GLEAM and ERA5 Land, to obtain an estimate of the observed FR,
indicated by the vertical lines. The shading around the lines shows the
range of one standard deviation from the mean trend, obtained by
bootstrapping the years that are included in the trend estimation.

The ETx7d forced trend from ERA5 Land lies at the very upper end
of the unforced trends and within the lower portion of the forced
trends. The forced trend for GLEAM is larger and lies clearly outside
the null distribution and well within the historical forced trends. For
the Northern hemisphere JJA ET, the distributions look very similar.
The trends of both observational products would be very unlikely in a
preindustrial climate but well within the expected trends under his-
torical greenhouse gas and aerosol emissions. It is also worth noting
that we find a similarly robust signal of change in JJA ET and ETx7d. The
observational products fall into similar quantiles of the historical and
PiControl trend distribution for both JJA ET and ETx7d. The ETx7d
trends are larger, but so is the range in the PiControl distribution,
which reduces the signal-to-noise ratio. The observational trends of JJA
ET are generally smaller, but so are the likely ranges of the PiControl
trends.

The observational summer trends (DJF) in the Southern Hemi-
sphere lie within the piControl distribution. However, the historical
CMIP6 simulations also show no clear trend in either direction. Thus,
we neither find nor expect a robust change in the Southern Hemi-
sphere average.

Additionally, we repeated the analysis for global and seasonal
mean ETx7d for the 2001 to 2021 time span, allowing the inclusion of
X-BASE (Fig. S7). We also find a strongly detectable signal in all
observational products for global and northern hemisphere JJA ETx7d,
with X-BASE showing the largest change over the time period. For
Southern hemisphere DJF ETx7d, we find no detectable changes.

## Regional changes
To investigate more regional changes, we focus on the SREX regions
defined in IPCC's AR6. We calculate the weighted spatial mean ETx7d
and JJA ET of all grid cells within a given SREX region and omit the
regularized regression step, as the regularization offers little benefit
over the spatial mean when estimating very regional FR. Thus, we
calculate the 1980-2023 Theil-Senn trends for both ERA5 Land and
GLEAM for ETx7d and JJA ET for all SREX regions over land (Fig. 4).
Since changes are less robust during Southern hemisphere summer,
we omit DJF ET from this analysis and only show it in the Supple-
mentary (Fig. S2). We repeat the procedure for forced and unforced
climate model simulations analogously to Fig. 3. To evaluate the con-
sistency of trends across datasets and assess significance, we perform a
series of tests for each region.

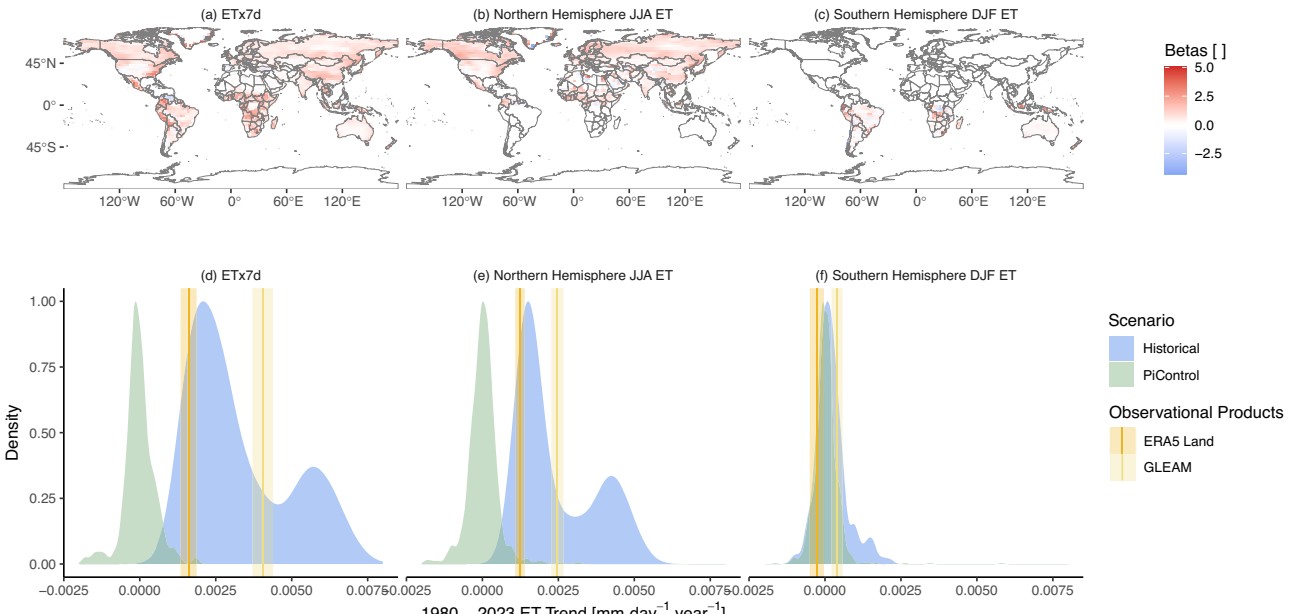

**Fig. 3 | Detecting changes in global ET.** Mapped coefficients for predicting the forced response of global mean ETx7d (**a**), Northern (**b**), and Southern (**c**) hemisphere summer mean ET. The coefficients have been scaled to the same variance but not centered, so positive values indicate positive correlation with the forced response. ETx7d (**d**), Northern hemisphere JJA ET (**e**), and Southern hemisphere DJF ET (**f**) forced trend density of historical (blue) and PiControl (green) climate model simulations between 1980 and 2023. Observational trend of GLEAM and ERA5 Land is indicated by the vertical lines, as well as the bootstrapped standard deviation by the shading around the lines. Maps made with Natural Earth.

Each hexagon is colored if the two observational datasets agree on the direction of change. This agreement occurs in slightly fewer than half of the regions for ETx7d. In contrast, for JJA ET, the observational data sets agree in most regions, with approximately two-thirds indicating an increase. Next, we test whether the two observational trends lie within the distribution of the historical simulations of CMIP6. Regions that pass this test are indicated with a circle, which is the case for almost every shaded region. Thus, observational products and climate models with historical emissions show trends that agree in many cases. We find significant trends (either outside the PiControl distribution or passing a Mann-Kendall test) for a number of regions. When a region passes both tests, it increases our confidence in the robustness of change in those regions. Note that an empty cell only indicates a lack of agreement on the observed trend sign, not agreement on the absence of a change. The strength of the coloring is determined by the number of tests passed, with stronger colors indicating increased confidence in significant changes.

A notable region is WCE, which passes all the tests for both ETx7d and JJA mean ET. Although the strong temperature trends over Europe have almost certainly helped to drive this trend, it is very likely to be superimposed on a trend of increased surface radiation over Europe in the last 40 years. This increase in surface radiation is known as global brightening driven by reduced aerosol emissions and concentrations as a result of air pollution measures[35]). Of course, there are also other processes that drive an increase in ET, such as greening. However, the regional strength of these changes does point towards a more regional driver, such as aerosol concentrations or dynamical trends, rather than a more global driver like temperature increase and greening. Regions in higher latitudes, such as NWN, NEU, RAR, and RFE, tend to show a clear tendency toward increases in both ETx7d and JJA ET. ET is expected to increase with increasing potential ET in regions with abundant water. The two regions that make up Northern South America also show increases in both ETx7d and JJA ET. Also worth noting are the JJA ET trends in Western North America, South-Eastern South America and South-Eastern Africa. The observed trends agree on a decrease, which is significant and outside the PiControl range but not

within the historical CMIP6 distribution. Assuming that the observed trends are reliable, climate models do not capture the strong observed decreases in these regions. When comparing the two maps, it is worth noting that all regions that show an increase in ETx7d also show an increase in JJA ET, with the exception of the Mediterranean. This agreement could indicate that the trends in the extremes are mostly driven by the trend in the mean, where changes in the whole distribution led to higher values at the tail end. However, ETx7d trends are larger in most regions than the ones for JJA ET in the observations (Fig. S3) and climate models (Fig. S4), indicating an additional increase in variability in the distribution of ET. For the regions where JJA ET decreased, we universally find no robust change in ETx7d. This is likely caused by long-term decreases in water availability, which are less visible when focusing on high ET extremes. However, a decrease in the mean and no change in the extreme suggest an increase in the span of the distribution, as ETx7d moves further from the seasonal mean. A greater variance in precipitation and ET is the core process driving drought risk[8]. The divergence of mean and extreme ET can again be seen in Figs. S3 and S4, where regions with decreasing JJA ET show less change in ETx7d. In general, we can see the impact of anthropogenic climate change on the hydrological cycle even in ET products, and changes can be detected on a hemispheric and, in some cases, even regional scale. Repeating the analysis for the 2001 to 2021 period with the inclusion of X-BASE does not lead to new insights. The short trend estimation period in combination with the large internal variability on regional scale does not allow for the detection of robust trends (Fig. S8).

### Record setting
We further highlight the strong changes in ETx7d in Europe by illustrating the year in which the record of a grid cell was set (Fig. 5).

All investigated observational products agree that a large number of grid cells have experienced record-setting years in the recent past. It is important to mention that because each observational product varies in length, the time period examined differs for each product. Nonetheless, all three agree on the prevalence of recent records across

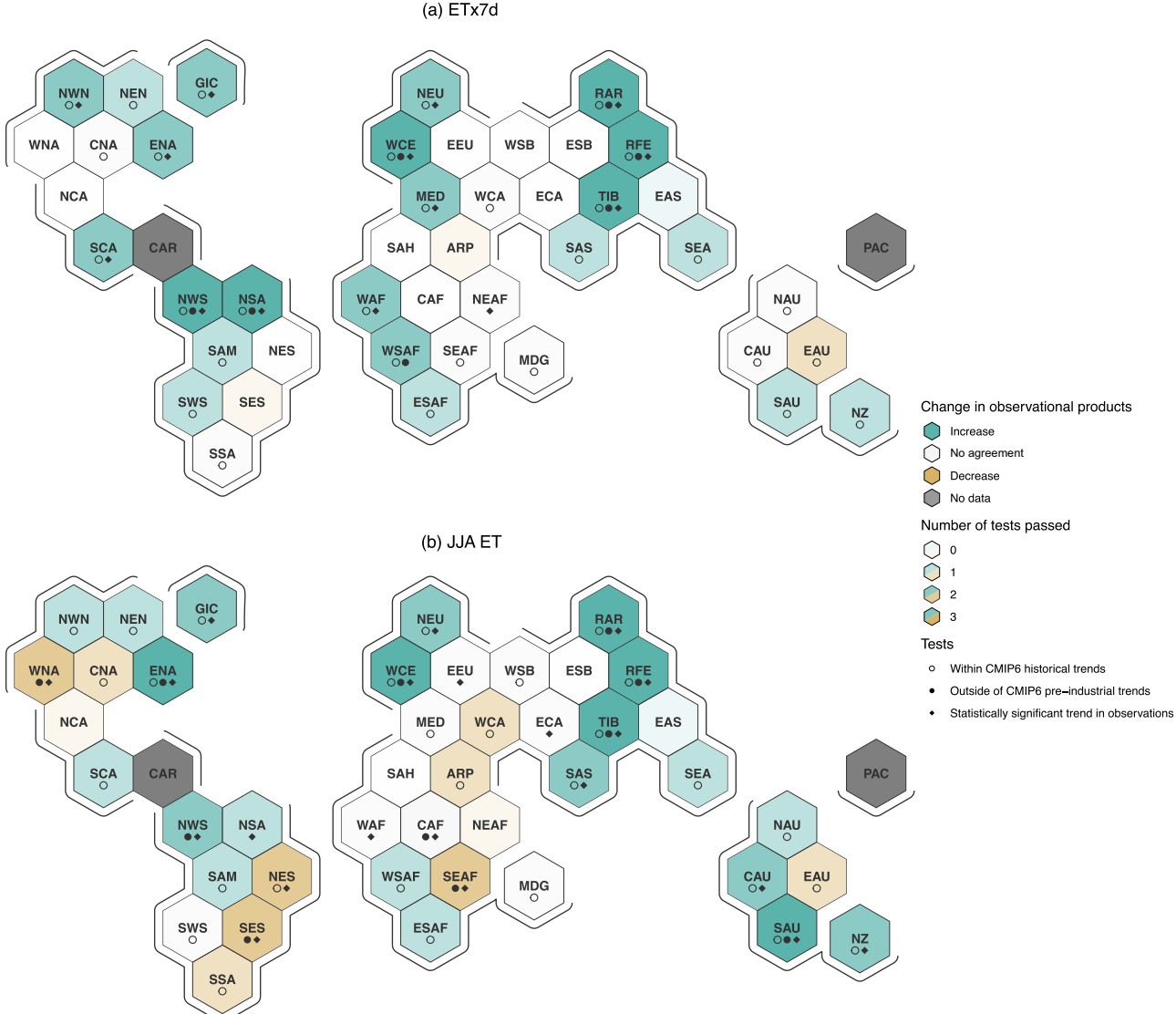

**Fig. 4 | Detecting changes in regional ET.** ETx7d trends are shown in (**a**) and JJA ET trends in (**b**) across all SREX regions over land. The color indicates regions where ERA5 Land and GLEAM agree on the sign of the 1980–2023 trend. The strength of the color indicates the number of the three tests a region passes. When these two trends lie within the distribution of CMIP6 historical trends, a circle is added to the region. When both observational trends lie outside the CMIP6 pre-industrial distribution, the black dot is added. Finally, when the observed trends both pass a Mann-Kendall significance test, the diamond is added.

the regions, with the exception being ERA5 Land in NEU, where a significant portion of records date back to around 1970. We compare the observational products with three climate models, which were selected due to their differing responses to climate change. These climate models agree with the observation in a skew of the record distribution towards more recent years. A notable exception is CESM2 in WCE, which shows a more balanced distribution. In the case of CESM2, there are more records set between 1970 and 1990, whereas ACCESS-CM2 shows a more bimodal distribution with high numbers of records at the beginning and the end of the 1950–2020 period. Overall, we conclude that, in many cases, climate models and observations agree on a tendency towards more recent record-setting events. This highlights that exceptionally high ET extremes over Europe have become more likely in the recent past, and this evolution will likely continue into the near future.

## Discussion

We examined extremes of high ET using both climate models and observational data. These events are primarily driven by warm spells with substantial atmospheric moisture demand and high incoming radiation, but are contingent on sufficient water availability. Rising temperatures contribute to higher atmospheric demand, allowing for more rapid evaporation. We study the seven days with the highest ET in a given year (ETx7d) and JJA mean ET. We detect trends in two observational products (ERA5 Land and GLEAM) in both ETx7d and northern hemisphere JJA ET, which would be very unlikely (<10%) to exceptionally unlikely (<1%)[36] without anthropogenically induced climate change. We further study regional changes in both ETx7d and JJA ET and find many regions that have experienced great changes over the last 44 years, which would also be very unlikely without human-induced climate change. Many climate models also indicate an increase in ETx7d over the past four decades, often in line with trends found in the observational products examined. For the majority of regions where we observe an increase in the extremes, we also find an increase in JJA mean ET. This alignment suggests that the trends in the extremes are partially driven by an upward shift in the total distribution. However, ETx7d trends increase more quickly than JJA ET, showing that there is an additional increase in variability, like it is

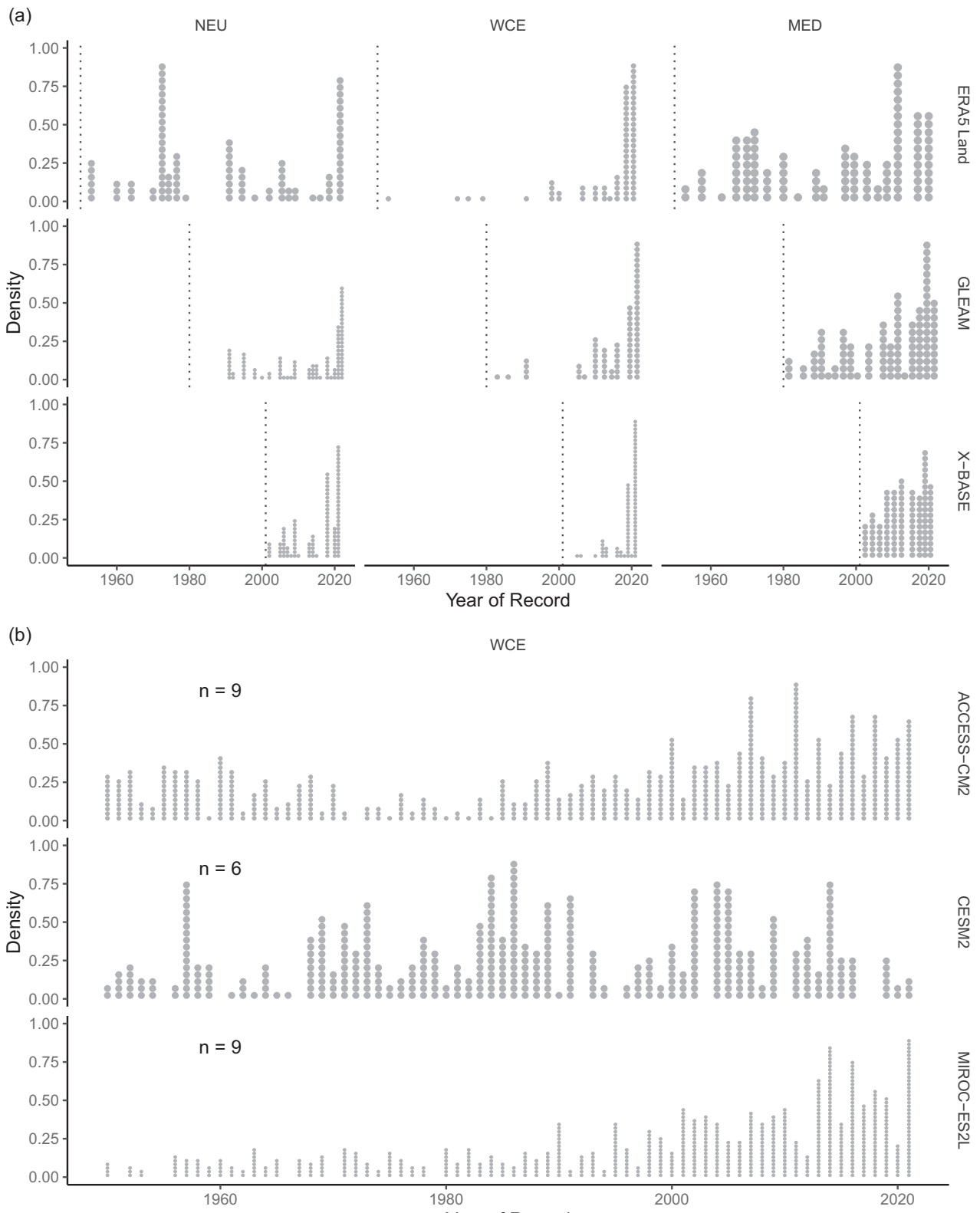

**Fig. 5 | Distribution of the years where the record ETx7d was set for all European grid cells.** Panel **a** shows three observational products (ERA5 Land, GLEAM, X-BASE), and **b** three climate models (ACCESS-CM2, CESM2, MIROC-ES2L). A point represents a grid cells's record being set in a given year. Europe is split up into the three regions Northern Europe (NEU), Western Central Europe (WCE), and Northern Europe (NEU). For each climate model, the number (*n*) ensemble members used in the analysis is noted in the panel.

commonly found in research on precipitation[6,37]. In regions where there is a decrease in the JJA mean, we do not see clear trends in ETx7d. The decreases in JJA ET are likely caused by water limitation as found by refs. 17,[18], but this limitation may not (yet) be strong enough to result in a detectable decrease in the extremes. By definition, extreme ET corresponds to periods in which conditions for ET are optimal in a given region. In a dry region, this can mean that a certain amount of water must be present for an ET extreme to occur, but we found no evidence for systematic decreases in extreme ET. In summary, for many regions, we find an increase in ET variability as the mean and the extreme move further apart, widening the distribution. This is further evidence for an amplified seasonal range in ET, which is a central driver for drought risk[8].

The global increase in temperature is a likely driver of increased ET, likely via the widespread increase in potential ET. However, if increases in global temperature were the most important factor, changes in ET should be more widespread, and their spatial distribution should reflect the surface warming pattern. Instead, we find multiple regional clusters, where historical ET trends seem especially strong. There are various other processes that explain these pronounced regional differences. Quantifying the contribution of these processes is outside the scope of this study, as these contributions vary regionally. However, we can offer some hypotheses about which processes are likely relevant.

Increases in vegetation cover, known as greening, have been found to increase ET globally[38], which would also lead to increases in extremes. Changes in radiation due to aerosols are a critical factor that affects ET[39–41], which is further exacerbated when studying extreme ET, since radiation sets an upper bound to ET. Reductions in aerosol emissions increase incoming short-wave radiation, thus raising this upper bound. In Europe, we found strong positive ET trends in both extremes and the summer mean. This region experienced substantial aerosol-induced dimming from 1950 to 1980, followed by brightening as emissions decreased[35]. Our trend analysis, beginning in 1980, identifies the associated increase in ET. If this is simply a return to the levels common before 1950, a continuation of these trends is not likely. This is suggested by multiple climate models that we evaluated in this study, but investigating this phenomenon in observations is not feasible, as the observational basis of ET before 1950 becomes very thin. However, when considering the magnitude of the recent observational changes, it becomes unlikely that they have been driven by changes in radiation only, but most likely a combination of effects. Furthermore,[42] found dynamically induced trends in European heat extremes, driven by changes in circulation patterns. It is plausible that more frequent heat waves would also increase potential ET and exacerbate the trend of mean and extreme ET in this region.

Representing aerosols, dynamical changes, soil water availability, and vegetation dynamics in climate models is a major challenge and is linked to great uncertainties. Improving this representation and understanding of these processes and their effects on the hydrological cycle should lead to a better understanding of the future risk of drought. Although it is beyond the scope of this study to officially attribute these trends to individual drivers, available evidence indicates that the changes in ET detected in this study are due to a combination of several factors, namely, the increase in evaporative demand, changes in vegetation cover and water availability, recovery from the aerosol effect and the role of unforced variability. The continuous effort to improve observational products is vital for studies such as this one, as the trends within the products used in this study are subject to substantial uncertainty. We attempted to navigate these uncertainties by checking whether the observational products agree with each other. Still, the trends within them should be interpreted with caution.

Our findings are in line with the increasing number of studies on anthropogenically forced changes in the water cycle[15,19,43]. Our findings

also give more support to recently found changes in flash drought frequency[44], by indicating regions where anthropogenic activity has measurably increased high ET extremes. Regions with strong ETx7d trends and regions indicated to have a higher frequency of flash drought[45] do not always overlap, showing that increased short-term ET is not the only factor driving the changes in these events. Nevertheless, we expect regions with strong ETx7d trends to transition more rapidly from wet to dry conditions, speeding up the development of flash droughts.

## Methods
### CMIP6
We use climate model simulations from the CMIP6[46] archive, which is conservatively regridded to a common 2.5 ° grid. The selection of climate models and their number of members can be seen in Table S1. We apply a 7-day running sum and extract the maximum 7-day ET for every year. This yields one value per grid cell and year of extreme ET, for which we will use the abbreviation ETx7d. We then calculate anomalies with respect to 1850–1950 from the historical simulation to account for differences in the mean state of the climate models, which simplifies fitting the regression model. This procedure is repeated for every ensemble member of a climate model available with the historical and SSP5-8.5 forcing scenarios[47]. The procedure results in 217 climate model runs from 23 different climate models. Climate models that exclude human influences (piControl) are utilized as a counterfactual to investigate potential trend ranges driven solely by natural variability. The piControl runs are continuous simulations over hundreds of years. Daily ET was recorded and available as a variable for a subset of 15 models, and those continuous simulations were split into chunks of 44 years. We then calculated the trend over all the chunks, resulting in 130 trend estimates.

### Observational products
In addition, we use three observational products. First, ERA5 Land reanalysis[48], which is a land surface model driven by meteorological variables from ERA5. Since it is an offline model, it does not represent land-atmosphere coupling as well as ERA5 itself. However, it is better suited to long-term trend analysis than ERA5 itself, as it represents soil water storage more accurately. ERA5 Land provides data from 1950, and we include data up to and including 2023. The original spatial resolution is 0.25°, but, like the CMIP6 data, it is regridded to 2.5°. The temporal resolution is 1 h, which we process to daily data before proceeding with the extraction of ETx7d or seasonal means. Second, we use ET from the Global Land Evaporation Amsterdam Model Version 4.2a (GLEAM)[49]. This product utilizes remotely sensed variables to drive a physical-statistical model hybrid balancing evaporative demand using Penman's equations[50], evaporative stress and water availability. Data are available from 1980 to 2023. The spatial resolution is 0.1° before regridding. The temporal resolution is daily for the extraction of ETx7d and monthly for the seasonal means. As an additional data set, we use X-BASE[51]. This is a new iteration of the popular FLUXCOM product, which uses machine learning to extrapolate tower-based eddy-covariance observations. Since it only provides data spanning 2001 to 2021, it is not used for trend analysis in the main text. The corresponding trend analysis from 2001 to 2021 is visible in the Supplementary (Figs. S7 and S8). The spatial resolution is 0.25° before regridding to 2.5°, and the temporal resolution is daily. Since the base period for the anomaly calculation (1850-1950) is not available for the observations, we instead used the whole observational period as a baseline. However, this has no effect on the trend within these datasets, which this analysis focuses on and is merely done to simplify fitting the regression model. To assess whether trends in the extremes are stronger than in the mean state, we also include seasonal mean ET for both December, January, February (DJF) and June, July, August (JJA) into the analysis. To obtain an overview of the meteorological situation

associated with such events, we extract daily values of various variables 10 days before and after the maximum ET is reached in ERA5 Land.

### Regularized linear regression

To study changes in seasonal and extreme ET, we use a detection and attribution framework. The overall goal is to isolate the effect of external forcing in the climate model (forced response) while minimizing the effect of spurious trends caused by natural variability. For this, we use regularized linear regression (sometimes referred to as ridge regression), which has been used successfully to detect changes in the daily temperature distribution[27] and in the mean annual and extreme precipitation[26]. The regression model is fitted to climate model data, where yearly ETx7d maps predict the FR in ETx7d of a given climate model. In this context, the global mean ensemble mean serves as a proxy for the forced response. Thus, every grid cell is a linear predictor of the global ensemble mean ETx7d.

The regression model can be represented as shown in equation (1). The FR is a multiplication of a vector containing all ETx7d values ($\mathbf{X}$) and the coefficient vector ($\beta$), which both have a length corresponding to the number of grid cells. The coefficient vector maps the contribution of every grid cell to the FR. The regression model, therefore, formulates a spatial weighting for approximating the ensemble mean from a given yearly map by avoiding regions of high internal variability.

$$FR_{year} = \mathbf{X}_{year}\beta + \epsilon \tag{1}$$

$\beta$ is found through the cost function (Eq. (2)), which minimizes the residual sum of squares, like in ordinary least squares regression. In addition, it minimizes a second term, which grows with the size of the coefficients, thus punishing the regression model for fitting large coefficients. Regularization can be particularly useful when dealing with correlated predictors, as is the case here. The balance between the two penalty terms is given by $\lambda$, which is found through cross-validation.

$$argmin\, RSS + \lambda \sum_{j=1}^{p} \beta_j^2 \tag{2}$$

The regression model is fitted across a variety of climate models to ensure robustness across a variety of distributions, guarding against overfitting to a single climate model. We select the optimal regularization as the best-performing regression model in a "climate model group as truth" cross-validation framework. The grouping is based on climate models that share similar architecture and cannot be considered independent of each other[52]. The "climate model as truth validation" emulates the distributional shift that occurs when applying the regression model to observations, leading to a more conservative regression model overall, which generally predicts smaller values and smaller trends. Once the regression model is fitted and the optimal regularization is chosen, we estimate the FR for all climate model members for the historical and preindustrial scenarios, as well as the two observational products, ERA5-Land and GLEAM. We calculate the 1980–2023 linear trends in the time series of all the resulting FR estimates. This procedure is repeated for seasonal mean ET.

### Regional analysis

We use the updated SREX regions from the 6th IPCC assessment report[53], which group climatically similar regions to explore the changes in ETx7d on a more regional scale. We first extract ETx7d on a grid cell level and subsequently calculate the weighted spatial mean. Thus, not all averaged values in a region have to stem from the same event. The regions are visualized in Fig. S1. To assess the changes, we simply fit regional Theil-Sen trends to observations and CMIP6 model simulations. While it would be possible to also employ the previously

used regression approach for this, we decided against it, as there are no established cases for extracting the regional FR using this methodology. We then calculate the probability of finding trends with the observed magnitude in either the historical or the PiControl simulations.

In order to test the robustness of the trends, we perform a series of tests that combine the three main lines of evidence for climate change. Understanding of physical processes, observational agreement, and considering the range of trends possible under natural variability. First, we check whether two observational products agree on the sign of the trend (Eq. (3)). Second, we test whether these two trends lie within the CMIP6 simulations of the historical climate to see whether the physical processes represented in the CMIP6 models can produce the observed trends. This is done by finding the quantile in the historical CMIP6 trend distribution to which the observational trend corresponds. If the quantile of observed trends is greater than 0.025 and smaller than 0.975, we consider them to be within the historical CMIP6 distribution (Eq. (4)). In the same way, we test whether the quantile of the observed trends is outside the likely range of PiControl trends (Eq. (5)). Finally, we test whether the observational trends pass a Mann-Kendall significance test with a $p$-value smaller than 0.05 (Eq. (6)). Both observational data sets must meet the test criteria for it to be considered a pass. The idea of combining the two significance tests in Eqs. (5) and (4) are to ensure robustness. The Null distribution of Eq. (5) should more accurately capture low-frequency variability, as it stems from fully coupled climate models. However, this Null distribution could also be artificially narrow due to climate model shortcomings, and we therefore add the well-established Mann-Kendall test.

$$sign(Trend_{ERASLand}) \equiv sign(Trend_{GLEAM}) \tag{3}$$

$$0.025 < q_{obs}^{Historical} < 0.975 \tag{4}$$

$$q_{obs}^{piControl} < 0.025 \lor q_{obs}^{piControl} > 0.975 \tag{5}$$

$$p_{obs}^{Mann-Kendall} < 0.05 \tag{6}$$

### Data availability

We used CMIP6 simulations accessible at https://esgf-node.ipsl.upmc.fr/search/cmip6-ipsl/. The observational products used can be accessed through the following links. The code used to process the data sets can be found in the code availability section. ERA5 Land: https://cds.climate.copernicus.eu/datasets/derived-era5-land-daily-statistics?tab=download GLEAM 4.2: https://www.gleam.eu/#downloads X-BASE: https://meta.icos-cp.eu/collections/_l85vWiIV81AifoxCkty50YI.

### Code availability

The code and the R environment used for the analysis and to generate the figures in the text can be accessed through https://doi.org/10.5281/zenodo.17092769. The R environment contains the data used to generate the figures from the main text.

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

## Acknowledgements

S.S. and R.K. acknowledge the project 'Constraints on near-term warming projections via distributionally robust statistical and machine learning' (COPE; grant agreement C22-02), funded by the Swiss Data Science Center. The authors thank Urs Beyerle, Lukas Brunner, and Ruth Lorenz for the maintenance of the CMIP6 data on ETH servers. We also thank all providers and maintainers of the observational data sets.

## Author contributions

All authors contributed to the conceptualization of the study. The methodology was developed by M.E., S.S., and V.H. under the supervision of R.K. M.E. conducted the formal analysis and writing of the initial draft. All authors contributed to the writing and review of the manuscript.

## Funding

## Competing interests

The authors declare no competing interests.

## Additional information

**Supplementary information** The online version contains Supplementary material available at https://doi.org/10.1038/s41467-025-67748-8.

