## [Transparent Peer Review file · Nature Communications]

Detecting Anthropogenically Induced Changes in Extreme and Seasonal Evapotranspiration Observations

Corresponding Author: Mr Marius Egli

Version 0:

Reviewer comments:

Reviewer #1

(Remarks to the Author)

Summary: the authors present an analysis of trends in extreme evapotranspiration (ET), diagnosed as the seven day maximum ET in each year. The study makes use of reanalysis and hybrid observed/model products in combination with GCM experiments drawn from CMIP6 in order to diagnose and explain trends at regional scale. I enjoyed reading the manuscript and feel it will be a useful contribution to the literature. Statistical methods used or trend attribution are clearly explained and, as far as I am able to judge, appear to be appropriately applied. I have a few comments for the authors to consider.

Specific comments:

1. The study is motivated in part by the concept of flash droughts, with the rationale that flash droughts are associated with periods of elevated ET that rapidly draws down soil moisture. I appreciate this framing and agree that there is sometimes a relationship between flash drought and ET extremes. But we shouldn't always expect the two to coincide. Indeed, as the authors note in the final sentence of the paper, there are regions of increasing flash drought that show no trend in ET extremes. I would suggest that this is because not all flash drought phenomena should be expected to be associated with elevated ET in the manner the authors have defined it: sometimes a diagnosed flash drought can come when soil moisture is already somewhat depleted, such that the rapid drying during the flash drought wouldn't show up as the highest ET of the year. This is described to some extent in a flash drought classification study of Osman et al. (2022)

Osman, M., Zaitchik, B. F., Badr, H. S., Otkin, J., Zhong, Y., Lorenz, D., ... & Holmes, T. (2022). Diagnostic classification of flash drought events reveals distinct classes of forcings and impacts. *Journal of Hydrometeorology*.

2a. The authors explain their choice of "observation" datasets quite clearly, and I can't disagree with their decision. "Purer" observation datasets from long-term flux towers or diagnostic satellite methods have limitations that would hinder their use for this kind of long-term, large-scale trend analysis. Nonetheless, I would appreciate it if the authors could be careful in their use of the term "observations" when referring to ERA5-Land (an offline model) or GLEAM (a hybrid product that depends in large part on a model). The authors do this to some extent when introducing the datasets, but an explicit discussion of the limitations of these datasets for trend analysis (always a challenge in advanced reanalysis products) or as fully reliable indicators of actual ET would be useful, perhaps in the conclusions and outlook section.

2b. Also on the topic of "observations": X-BASE, and its underlying input datasets, is the closest thing the analysis has to a direct observation of ET. The authors state that they do not use it in analysis because it only has data "for 2001" (Section 2), but I think they mean "from 2001." While that's not an ideal length of record for trend analysis, it's not impossible to calculate a trend from such a record. It might be interesting to do so and to mention the results where useful when interpreting trends seen in GCMs and in GLEAM and ERA5-Land.

3. Resampling to 2.5 degree resolution is, as the authors indicate, conservative, and is a reasonable way to perform the study. But it also means that the analyses are looking at ET averaged across very large areas that include diverse land

cover types and vegetation conditions. While it would be difficult or impossible to examine sensitivities to sub-grid scale vegetation in CMIP6, it should be possible to consider land cover sensitivities in ERA5-Land and GLEAM. Are the trends, in the reanalysis/hybrid products, being driven by particular land cover types? This could inform our interpretation of trend results and is relevant to the flash drought framing. While flash droughts can occur in any land cover type, flash drought processes are expected to be quite distinct in, for example, forests vs. croplands. Stratification by land cover type might also be relevant to the authors' comparison of trends in extreme ET vs. trends in seasonal JJA ET.

4. I have mixed feelings about the jump from Europe to global in the results section. On the one hand, it's obviously useful to have hemispheric and global analyses as a contribution to the literature, and the use of IPCC regions and figure types will be helpful for future assessment reports. But the global analyses raise so many questions. All of the process-informed work presented in the manuscript is for Europe. Leaping to global, we see an interesting distribution in trends in Figure 3 and mixed results across regions and climate zones in Figures 3 and 4. It feels unsatisfying to present these results without explanation. Practically every part of the world shows some kind of interesting combination of results--what's going on in the Amazon? The Sahel? Australia?--and it's not obvious that the explanations provided for Europe apply in these other places. So . . . on the balance of it, my instinct would be to turn this into a longer paper or two separate papers. Either dig into the global results in earnest, or present this first, short paper as an introduction of the method with application to Europe and then write another (potentially very high impact paper) that goes global in earnest. I'm not going to insist on that, as there's nothing that I can see that's "wrong" in the global results. It would just be my recommendation.

Reviewer #2

(Remarks to the Author)

Review of "Detecting Anthropogenically Induced Changes in Extreme and Seasonal Evapotranspiration Observations"

Summary and key results

Motivated by the contribution of high evapotranspiration events to flash droughts, this manuscript investigates anthropogenically induced trends in extreme and summer mean evapotranspiration (ET) on a global as well as regional scale. The study is based on both climate model outputs from CMIP6 (piControl/historical/SSP5-8.5), used to construct ET trend distributions in a world with and without global warming, and three observationally-informed model products (ERA5, GLEAM, X-BASE). A Ridge regression model is used to estimate the forced ET trends. Comparing the observational trends to the historical and piControl trend distributions reveals robust positive trends in both extreme and northern hemispheric summer ET which are very likely induced by anthropogenic activity. In addition, the study analyses ET trends for the 45 SREX regions. A number of regions in Eurasia and South America exhibit robust positive extreme and JJA ET trends. The authors explain positive trends as a mix of rising temperatures, leading to an increase in vapour pressure deficit, and an increase in incoming shortwave radiation, caused by aerosol reductions in recent decades.

General Assessment

With its focus on extreme events, the presented analysis makes a relevant, significant, and novel contribution to our understanding of how ET — a central component of Earth's hydrological cycle — evolves under the influence of global warming. Especially the use of a sophisticated regression technique designed to minimise the notoriously confounding influence of natural variability adds to the novelty and quality of the work. While the statistical method seems sound, and the results are compelling, I am less convinced by several aspects of the interpretation of the results, and especially the claimed link to flash droughts, for which I feel the manuscript lacks supporting evidence. In terms of style, the manuscript is written in appropriate language and generally good English. Nevertheless, I found it difficult to follow on my initial read-through, in parts because of occasional contradictory statements that impede the flow of the argument. The text would benefit from a more effective use of paragraphs (or even subsections) to structure ideas. In some places, essential information appears to be missing. I provide more details on these and other concerns/suggestions in the following major and minor comments. Where applicable, I reference the corresponding line numbers which can be found in the attached version of the manuscript.

Major Comments

1. Assumption that ET_{7d} occurs when the soil is saturated and PET is met

The maximum value of 7-day accumulated ET at a given location in a given year does not automatically imply a situation in which $ET = PET$. There are presumably many places, especially in arid and semi-arid regions, where $ET < PET$ all year around. Also the typical soil moisture values of 0.35 in Fig. 1g seem to point at some moisture limitations. Whether the $ET = PET$ assumption indeed holds could perhaps be checked with a quantitative analysis. Alternatively, if the authors consider $ET = PET$ to be an important condition for their objectives (e.g. for removing water limitation as a confounding variable (line 60), or for testing if projected PET increases manifest in ET increases (line 63+64)), then an additional filter for $ET = PET$ (or approximately equal to) could be applied to the data. I am however not convinced of the importance of the $ET = PET$ assumption, given my next comment.

2. Link between extreme ET, PET, VPD and flash droughts

It would be useful to provide a definition of (flash) droughts in terms of the variable by which they are identified. If relatively dry soils are a typical drought characteristic, as would be consistent with the claimed role of high ET in driving flash droughts, then I would expect that particularly the moisture-limited regions around the world are prone to experience flash droughts. In contrast, a full soil moisture storage should reduce the drought risk. If this reasoning is sound, then locations and times where $ET = PET$ are precisely not the ones that are particularly relevant for flash drought occurrences. Thus, before concluding which regions are allegedly more drought-prone due to a positive ET_{x7d} trend, one might want to add an additional check of the regional soil moisture conditions. After all, a positive ET_{x7d} trend could also originate from a positive soil moisture trend, e.g. due to increased precipitation. I also got generally a bit confused by the discussion of the relationship between ET, PET, and VPD (and their link to flash droughts). What is the exact role of VPD in contrast to PET? In lines 29 to 36, the authors state that many flash droughts are driven by high PET, and then assuming $ET = PET$ during ET_{x7d} events links trends in ET to trends in PET. But the physical explanation for these trends in the Conclusions section only involves VPD. I think it would be helpful to clarify the various relationships between variables by presenting the equations that link them. Given the non-straightforward link between ET_{x7d} and flash droughts (and the fact that no drought-specific analysis is performed), I want to point out that the investigation of anthropogenically-induced trends in ET variability and extremes is relevant for numerous atmospheric and land climate research questions, and therefore has great value by itself. It might be worth contemplating to assign flash droughts a less prominent role in the manuscript, e.g. by mentioning them as one phenomenon typically associated with ET extremes.

3. Presentation of 'observational data' and missing information in data section

Except for X-BASE, none of the presented observational data provides information from direct ET measurements, and even in the case of X-BASE, spatial coverage is achieved by extrapolation using a ML model. (As such, it is also misleading that line 46 claims that we have accurate data since the late 20th century.) While I understand the use of these datasets as our 'best guess' of real ET, I think the text should be more transparent about their true nature, i.e. them being model products. Labelling them as 'observational' is fine as long as it is made clear that this is a simplification made for convenience. Beyond the transparency, I wonder if the limitations of these data sets have any implications for the results and conclusions. Perhaps the authors could comment on any relevant known biases. Lastly, I was missing some essential information about the observational dataset in the data section, namely: spatial and temporal coverage and resolution.

4. Insufficient discussion of limitations inherent to methodology

The second paragraph of the Introduction dwells on how difficult it is to know anything about ET. Listing the different problems at this point raises the expectation that the manuscript is going to present solutions to all these problems. However, the only actual 'solution' that it presents, is the use of a regression technique that tackles the confounding influence of natural variability. It would be more informative to limit the discussion of problems in the Introduction to those which the paper actually tackles, i.e. the extraction of the anthropogenic signals from natural variability and the use of ensembles and more than one observational dataset to limit the detrimental effect of model biases and other limitations. A more comprehensive discussion of remaining limitations could be moved to the data and methods or results section.

5. Generalisation of European context

I was surprised to find such a strong focus on central Europe, given that Figure 1b in Christian et al. (2021) shows that the flash drought occurrence in Europe is rather low. Other regions with significant ET_{x7d} trends, such as the northern part of South America, are known flash drought hot spots, rendering them potentially more interesting. This makes me wonder if the European focus was primarily chosen due to the authors' affiliation with European institutions. Should there be other reasons to only zoom in on Europe, they should be made explicit. Another issue with the Euro-centric view is that drivers of ET extremes might differ in different regions. The manuscript identifies heatwaves as the prevalent meteorological background of ET_{x7d} . However, with their connection to stationary Rossby waves and suppressed convection, mid-latitude heatwaves may present themselves differently than tropical heatwaves. Similarly, the suggested role and timing of aerosol emissions seems specific to WCE. Aerosol emissions (or their reductions) might follow a different timing in other parts of the world which should be reflected in the respective ET_{x7d} behaviour if the proposed mechanism is to be considered robust. Regarding the influence of aerosols, I stumbled over this paper by Zou & Yang et al. (2025) who find that aerosol reductions over China increase extreme precipitation which may have implications for ET_{x7d} in the region. (<https://doi.org/10.1029/2024GL113887>) In summary: To which extent can the factors identified as 'drivers' of ET_{x7d} be generalised? Can the regions showing significant anthropogenically-induced ET_{x7d} trends be discussed in a joint framework?

6. Discussion of future scenario inconclusive and lacks quantitative analysis

I assume that the future SSP5-8.5 scenario was included to get a feeling for whether identified historical trends should be expected to continue into the future. However, given the strong model disagreement and regional dependence of future ET_{x7d} , I am not sure what to take away from this discussion. Further, the claim that water limitations drive a decrease in ET_{x7d} towards the end of the century (line 168 + 207) is not backed up by a quantitative analysis. Perhaps the discussion of future ET_{x7d} could be dropped in favour of a more detailed explanation of the Ridge regression technique, which to me seems like a strong point of the work, as well as a more globally valid discussion of potential ET_{x7d} trend drivers?

Minor Comments

Line 7 + 8: Given that ET was already identified as a flash drought driver in other studies, it is perhaps rather the trend in

flash droughts that has not been looked at through the lens of ET?

Line 9: what is the novelty regarding flash droughts? Did the study identify new regions that are at flash drought risk? Or does the lack of clear ETx7d trends in many of the flash drought prone regions around the world mean good news for these regions?

Line 14: That the sign of ET trends depends on water supply and demand is a very generic statement. Apart from that, the evolution of supply and demand were not studied and compared to the trends.

Line 14+15: universal increase in ETx7d is not true, see the many white tiles in Fig. 4a as well as statements in line 262 + 263. Further, claim of water demand as the reason for the increases is just a claim without proof.

Line 29 + 30: Not sure about this presentation of cause and effect (although I admit that a discussion of circular mechanisms is always challenging). My attempt: Increasing temperature T increases the atmosphere's water holding capacity by raising the saturation vapour pressure q_s . Rising T and q_s together lead to an increase in PET. By how much VPD increases (or RH decreases) depends on both q_s and the response of ET to the altered atmospheric conditions (when the soil moisture is abundant, it will increase less than if the soil is moisture-limited). This is where equations for PET and VPD would come in handy.

Line 31: Really about 50% while the other 50% is due to a lack of precipitation. Perhaps this should be noted.

Line 39: Why does the levelling trend occur, is it relevant to the present study?

Line 41: What is their definition of extreme ET, do you use it, why not? (Perhaps something to pick up on in the methods section)

Line 42 + 43: no study investigated 'trends in' seasonal and extreme ET globally? Given that this work also does not study seasonal and extreme ET in itself, this would make more sense.

Line 65: detectable increases in magnitude/frequency?

Line 66: missing word + what does "amplified onset" mean?

Line 68: Only CMIP6 is mentioned here but the central part of the manuscript is that it looks at "observations" and contrasts them with CMIP6, the latter providing a benchmark

Line 71: What is the motivation to look at seasonal ET? How is it relevant to flash droughts? + should be 'global and hemispheric' not 'global or hemispheric', or just 'global'

Line 75: short description of the different simulations missing (historical/piControl/SSP5-8.5, with time periods and temporal resolution). Temporal and spatial coverage and resolution are also lacking for the observational datasets.

Line 79: why normalisation if study anyway about trends? Does the 1850-1950 period come from historical or piControl runs?

Line 81: climate model 'simulations'

Line 85: contradicts the claim that we need not worry about soil moisture because ETx7d represents cases where soils are fully saturated anyway (which I doubt, see major comments). Apart from that, I would expect that the focus on extremes, as well as the invoked dominance of atmospheric demand, makes land-atmosphere interactions particularly important.

Line 91: X-BASE provides data 'since' (not 'for') 2001?

Line 94: why use a "wrong" compute baseline rather than no baseline for observational data?

Line 98: which variables? CMIP6 variables are not shown, however, this could be a nice additional analysis to show because it would corroborate the physical soundness of the simulations.

Line 102: remote 'and'

Line 103: effects 'of'

Line 106: Is the ensemble mean ETx7d also a global map? At some point, spatial averaging is needed to arrive at a trend number (unless each grid cell contributes a trend value). When does this averaging happen? Perhaps it would be generally helpful to provide more information on how the algorithm goes from a map of ETx7d to a map of coefficients, and from there to a number. A schematic or equation might help.

Line 109: What does "ensure robustness across a variety of distributions" mean?

Line 110: didn't understand, generally: naming consistency would help the reader, I found the following names that (I think) referred to the Ridge regression model throughout the text: model, statistical model, Ridge regression, regression model, linear model

Line 113: General advice on avoiding "This", especially at the beginning of sentences, because what it refers to is often ambiguous. Other instances of this problem in lines 118 (which procedure?), 160 (which period?), 201 (which events?), 348 (what?), 360 (what?), 361 (which phenomenon?)

Line 114: What does conservative model mean here? Does it provide a smaller range of trends than a less conservative version would?

Line 115: chosen or found? I didn't really understand what the regularisation is or does and this made me wonder if it is something to pick or whether it is determined by running some optimisation algorithm. Line 120-123: I would provide the information on why Ridge regression wasn't used again already here instead of in the results section.

Line 124: 'both' the historical and ... Line 124: I would stick to the short hand piControl rather than spelling out pre-industrial for the same kind of consistency as always calling the regression model by the same name.

Line 126: Rather than checking the physical understanding, Equation 2 checks for physical consistency. A small suggestion would be to swap the order of Equation 1 and 2 because then they would match the order of lines of evidence outlined here.

Line 130 + 131: Sentence hard to understand, my alternative suggestion: "This is done by finding the quantile in the historical CMIP6 trend distribution to which the observational trend corresponds."

Line 133: Equation 3

Line 152: Perhaps cite a heatwave review paper for this claim? E.g. Domeisen et al. (2023)

Line 160: Are aerosol effects and changes represented in all CMIP models? How do piControl and SSP5-8.5 handle aerosol emissions?

Line 164: I'm not convinced there is a clear increase in the observations for MED given the large variability of observational values

Figure 1: Given that there is space, I would spell out all variable names in the subplot titles for consistency (had to guess what SAT and thought of saturation at first before realising that it must be surface air temperature..). Soil moisture = relative soil moisture saturation?

Line 168 + 194: Can this water limitation be seen in the data? E.g. in soil moisture just before ETx7d which declines towards later years and more so for models with stronger negative anomaly? Why is water limitation not/less visible in MED and NEU? (Same issue in Line 345)

Line 189 (caption Fig. 2): 'ETx7d' intensity 'anomaly'

Line 192: perhaps better for the reader to know the dates that day 125 and 225 correspond to, this gives a better feeling for the ETx7d season.

Figure 2: the visibility of individual model spreads is in parts rather poor. I'm not sure what the best thing to do would be. If the journal allows for a bigger figure, one could try splitting each subplot in two and only show two to three models at a time. Alternatively, providing the model means as coloured lines might help to guide the eye? Or choosing different colours?

Line 199 + 200: An intensification is only really visible for GLEAM. X-BASE has a positive jump but preceded by a number of years with a negative trend, so not sure this is signal or noise. ERA5 Land line not visible at all.

Line 201: What is the motivation to look at the timing? Was there an initial expectation? What is the reader supposed to take away from this analysis?

Line 213: Shift in the distribution of ETx7d?

Line 226: 'This' (not 'That')

Line 227 - 229: "This results in" makes little sense in this context. Also just for clarification: does one end up with an ensemble of ensemble-mean estimates from the statistical model?

Line 232: coefficients are not generally positive, there are some negative ones, so perhaps better 'mostly positive'?

Figure 3 a - c: What is the meaning of the Beta values? Also, interesting bimodality in the historical distributions. Any idea where this comes from? Has it perhaps any connection to the Wet-get-wetter-dry-gets-drier paradigm? Caption: Line 217: obsolete colon, line 218: two c)

Line 250: What follows after “average.” was pretty difficult to grasp, had to read it several times. Perhaps this could be made more clear by not jumping back to Fig. 3 d + e but discussing this when first discussing these two subfigures figures?

Line 254: This would be natural place to start a new subsection

Line 255: mean of ETx7d and JJA ET

Line 267: do not disagree = agree

Line 268: worth naming these regions and perhaps contrasting them with Fig. 1 in Christian et al. (2021)? How does increase in ETx7d correlate with drought occurrence? Perhaps this manuscript says not only something about regions potentially at increased drought risk in the future but also something about already drought-prone regions for which a deterioration of the situation is not expected?

Line 278: Is it possible to show PET directly to corroborate the claim that ET trends follow PET trends? And how does the claim that ET increases with PET fit together with the statement in the introduction that PET and ET have been shown to evolve differently in a warmer climate?

Line 287: indication = indication? + How can a trend in the extremes be driven by trends in the mean? Extreme values enter the computation of the mean value, not the other way around. If anything, a trend in the extremes can drive a trend in the mean. I suspect what is meant by this statement is that the entire distribution is simply shifted to higher values. However, the next sentence already contradicts such a notion because extremes increase more strongly than the mean. (Same issue in Line 342)

Line 289: ‘Figure’ SI 4

Line 293: This is a trivial statement. If changes are detected on a larger scale then it is impossible not to have changes on a smaller scale since the smaller scale changes integrate to give the larger scale changes. The statement only works the other way around (from small to large scale): Only if regional scales cancel out on could end up with a situation in which there are only regional and no hemispheric changes.

Line 294: One 70-year record per cell of the historical simulations?

Figure 5 caption: First line is not a sentence, Line 338: regions, Line 339: two ‘in’ What is the bin size for records on the x-axis? Given the different dot sizes, the bin size seems to be different for different datasets. Why?

Section 4: Summaries and conclusions are typically written in past tense, at least for the bits that recapitulate what was done in the study.

Line 310: proof lacking for heatwaves being the primary driver in non-European cases

Line 313: Would summarise approach again: “By comparison with the trend distributions constructed from forced and unforced CMIP6 ensemble simulations, we detect...”

Line 314: How were these likelihoods determined?

Line 350: factors

Line 350 + 351: Surface warming is also not spatially homogeneous and VPD depends on the response of ET, horizontal moisture advection etc...

Line 353 - 356: Aerosols certainly play a role but there are many other factors that are likely important, too, such as soil moisture, precipitation changes, changes in wind and plant covers, ..

Line 360: Fig 2 does not show a simple return to 1950 levels of ET

Line 365: dynamical changes of what?

Line 372 + 373: Where were the effects of increased VPD visualised?

Line 373 - 375: Are the regions highlighted in this manuscript consistent with the spatial changes in flash drought frequency presented in Zheng et al. (2023)?

Line 376: what does “strengthened onset” mean? Earlier/more rapid/..?

Figures SI3 and SI4: It would be helpful if all axes would span the same ranges.

Reviewer #3

(Remarks to the Author)

This manuscript investigates long-running records of evapotranspiration to detect trends in extremes. The datasets include observation based records as well as climate model records. The modeled datasets further include a control dataset that is modeled for a world without human influence on the climate. The authors focus on the 7-day running total of ET, and specifically its maximum anomaly from that records long-term mean for a given region in each year in the dataset. This selection reduces the number of confounding factors that may influence ET. The authors detect trends in the two longer records of observational data that would be very unlikely to occur without the human induced climate change. The study gives some additional insight in regional trends and contribution factors. I find the methodologies and conclusions well supported and appropriate for publication after only minor changes.

Some more discussion would be appreciated of the disagreement in day of maximum ET between observations and models. The disagreements are fairly large, in the order of 1-2 months for the MED region. Can the authors provide more insight in what is happening here? From the presented data it is difficult to judge if the large differences are a result of only small changes in event ranking, or result from important methodological gaps between models and observations. Perhaps plotting the timing of the second highest 7-day period (that doesn't overlap) in the same plot could test if there is more agreement in the top-two periods than there is in just the maximum?

Further minor comments are included in the annotated manuscript.

Version 1:

Reviewer comments:

Reviewer #1

(Remarks to the Author)

The authors have addressed my concerns with the original manuscript, and I am pleased to recommend publication in present form.

(Remarks on code availability)

Reviewer #2

(Remarks to the Author)

The authors have addressed the vast majority of my comments in a careful, informative, and satisfactory manner. The manuscript now reads more clearly, is well-structured, inconsistencies have been removed, and I believe the claims and explanations are now sufficiently substantiated by the presented analyses. In fact, I think the manuscript in its revised form is a really great and impactful paper that will find wide interest in the community.

A number of minor comments/suggestions that remained or refer to newly added sections are listed below:

Abstract line 7: probably not an "increase in observational products" but rather "increase of evapotranspiration (ET) in observational products" ;)

Line 58 - 60: I still find this sentence to be quite generic with the exception of mentioning ET's role in the energy budget which is already mentioned in Line 19. While I won't insist on its removal (after all, it is not wrong), I don't see an added benefit in keeping it and I think this paragraph would just read as well or better without it.

Line 64+65: Why is water limitation portrayed as a confounding variable if soil drying is a direct consequence of anthropogenic warming in many parts of the world which is also used to explain some of the results later in the text. Wonder if this framing is a remnant from the initial narrative of the first draft where $ET = PET$ was assumed.

Line 76+77:

- global 'and' hemispheric (the study does both, not one or the other based on certain criteria)

- This sentence is a bit confusing: Is it meant to express that trends are detected in observations and robustness is tested by comparing with models? If so, my suggestion:

"We then apply methods from detection and attribution studies to find trends in extreme and seasonal ET on a global and hemispheric scale in observations, and test the robustness of these trends by comparing them to trends in CMIP6 model simulations."

Or alternatively:

"We then apply ... studies to both observational data and CMIP6 model output in order to find robust trends in extreme and seasonal ET on a global and hemispheric scale."

Line 86: facilitates or simplifies?

Line 90: Didn't understand the non-overlapping 44-year trends and where the 15 different climate models come from,

because two sentences earlier, the text mentions 23 different climate models.

Line 98: typo 'temporal'

Line 103: missing unit °

Line 109: not available 'for/from observations'

Line 115: in both ...? the ERA5 Land.

Line 136: singular or single?

Line 165 + 166: Perhaps add one more sentence here, explaining why this last test in Eqn. 6 is performed? (Also, I would still swap Eqn. 3 and 4 to match the order of tests mentioned in the text, but if they were kept in a different order on purpose, I won't insist on the change.)

Line 177: 'The two strongest ET events in the study region'

Line 181: "The two strongest events" —> couldn't tell if this was referring to the WCE region again or still to the US. For clarity, perhaps add '... events in the WCE region'

Line 184: 'suggesting' rather than indicating + the 'this' in the next sentence is a bit ambiguous. Either it refers to the sentence before or, because it is the last sentence of the paragraph, it refers to the entirety of observations described in this paragraph.

Line 186: Wonder if the switch to the temporal evolution and timing of ETx7d here warrants its own subsection. The way it is right now, the subsection title "What is an extreme ET event" no longer makes much sense beyond this line.

Line 206: The paragraph on the timing comes a bit unexpectedly/abruptly since timing has not been mentioned earlier as an objective or matter of speculation. The hypothesis of an earlier timing of extreme ET events could perhaps be added to the introduction (or alternatively a remark that, besides intensity, also timing of ETx7d could be affected by climate change).

Line 208 + 209: do you mean the tendency visible towards the end of the 21st century?

Line 255: typo "in for both"

Line 262-264: What signal does this refer to? This paragraph talks about the DJF trends where no strong signal was found... I'm confused.

Line 274 + 275: These two sentences make it seem like there is agreement about only half of the regions, but if I read Fig. 4 correctly, then no colouring doesn't necessarily mean disagreement but it could also mean an absence of any statistical trend, i.e. observations and models could agree on the fact that there is no trend. Is this correct?

Line 289+290: If this refers to RAR, RFE, and TIB which, like WCE, stand out as particularly clear cases of positive ETx7d and JJA ET trends, then I would mention these regions explicitly as examples.

Figure 5b: Why were those three climate models selected? Are they capture the range of models?

Line 325 + 326: were probably meant to be removed

Line 349: What does low frequency variability mean here, how is it linked to increased variability as seen in the difference between ETx7d and JJA ET trends?

Line 352 + 353: Not sure if the author's response to my previous comment comes out clearly enough in the text. My suggestion would be to replace "and evenly distributed" with ", and their spatial distribution should be reflective of the surface warming pattern."

Line 369: I wonder if "as in changes in circulation patterns" is too colloquial. I would reformulate to sth like: "Furthermore, Sing et al., 2023 found dynamically-induced trends in European heat extremes that are driven by changes in circulation patterns."

Line 71 + 386: I am perhaps being too picky here, but in my mind, the word "onset" refers to a point in time and, hence, cannot be speeded up, in the same way as it couldn't be strengthened or amplified. I would therefore rather use "speeding up the development of flash droughts". However, I think the context makes clear what the authors mean.

One last point: The very last sentence of the manuscript is rather "weak". It doesn't emphasise the scientific gain of the study, but rather mentions one more "caveat" of the presented results. If this sentence could be moved to the main discussion of the results, the paper would end on a much more powerful note which I think it deserves.

(Remarks on code availability)

Reviewer #1 (Remarks to the Author):

Summary: the authors present an analysis of trends in extreme evapotranspiration (ET), diagnosed as the seven day maximum ET in each year. The study makes use of reanalysis and hybrid observed/model products in combination with GCM experiments drawn from CMIP6 in order to diagnose and explain trends at regional scale. I enjoyed reading the manuscript and feel it will be a useful contribution to the literature. Statistical methods used or trend attribution are clearly explained and, as far as I am able to judge, appear to be appropriately applied. I have a few comments for the authors to consider.

We thank the reviewer for their positive evaluation and will highlight the adjustments made in the following comments.

Specific comments:

1. The study is motivated in part by the concept of flash droughts, with the rationale that flash droughts are associated with periods of elevated ET that rapidly draws down soil moisture. I appreciate this framing and agree that there is sometimes a relationship between flash drought and ET extremes. But we shouldn't always expect the two to coincide. Indeed, as the authors note in the final sentence of the paper, there are regions of increasing flash drought that show no trend in ET extremes. I would suggest that this is because not all flash drought phenomena should be expected to be associated with elevated ET in the manner the authors have defined it: sometimes a diagnosed flash drought can come when soil moisture is already somewhat depleted, such that the rapid drying during the flash drought wouldn't show up as the highest ET of the year. This is described to some extent in a flash drought classification study of Osman et al. (2022)

Osman, M., Zaitchik, B. F., Badr, H. S., Otkin, J., Zhong, Y., Lorenz, D., ... & Holmes, T. (2022). Diagnostic classification of flash drought events reveals distinct classes of forcings and impacts. *Journal of Hydrometeorology*.

We fully agree that a flash drought does not necessarily have to manifest itself as an ETx7d event and not every ETx7d event is a flash drought. However, increase in ETx7d would still likely coincide with more rapid drying in the case of a flash drought, thus we believe that the framing is still valid. Of course, there is variance on a case by case basis but our goal here is highlighting the overall direction. However, we have generally rephrased the link to flash drought, to be more in line with current flash drought literature.

2a. The authors explain their choice of "observation" datasets quite clearly, and I can't disagree with their decision. "Purer" observation datasets from long-term flux towers or diagnostic satellite methods have limitations that would hinder their use for this kind of long-term, large-scale trend analysis. Nonetheless, I would appreciate it if the authors could be careful in their use of the term "observations" when referring to ERA5-Land (an offline model) or GLEAM (a hybrid product that depends in large part on a model). The authors do this to some extent when introducing the datasets, but an explicit discussion of the limitations of these datasets for trend analysis (always a challenge in advanced reanalysis products) or as fully reliable indicators of actual ET would be useful, perhaps in the conclusions and outlook section.

This is a fair point. We have added several sentences to the introduction (lines 43-50) about some of the known biases and key shortcomings of these products. In addition, we have added the following sentences to the outlook.

"The continuous effort to improve observational products is vital for studies such as this one, as the trends within the products used in this study are subject to substantial uncertainty. We attempt to navigate these uncertainties by checking whether the observational products agree with each other. Still, the trends within them should be interpreted with caution."

In general, we have also adjusted the wording, moving away from the term "observations" in favor of "observational products".

2b. Also on the topic of "observations": X-BASE, and its underlying input datasets, is the closest thing the analysis has to a direct observation of ET. The authors state that they do not use it in analysis because it only has data "for 2001" (Section 2), but I think they mean "from 2001." While that's not an ideal length of record for trend analysis, it's not impossible to calculate a trend from such a record. It might be interesting to do so and to mention the results where useful when interpreting trends seen in GCMs and in GLEAM and ERA5-Land.

Absolutely, this was a typo. Thanks for pointing it out. We have repeated the analysis shown in figures 3 and 4 for the 2001 to 2021 period, now with the inclusion of X-BASE. The corresponding figures can be seen in the supplement (S17 and S18).

For ETx7d, we find that all three observational products produce a trend larger than the 0.98 quantile of CMIP6 PiControl simulations, with ERA5 Land having the smallest and X-BASE the largest trend. For JJA NH ET, the

observational products generally agree, however the trend within ERA5 Land falls into the 0.93 quantile of CMIP6 PiControl simulations and can thus not be considered statistically significant. The trends of the other two products fall into the 0.98 quantile.

For DJF SH ET we still do not find any significant changes.

For the regional analysis we find that all three products agree on the trends in a similar number of regions (Figure SI 8). However, we do not find any regions with statistically significant trends or trends outside the preindustrial trend distribution apart from ETx7d in WCE. This is a testament to the large internal variability on regional scale and that regional trends over a relatively short time period of time should be interpreted with care.

From this additional analysis we conclude that on a hemispheric scale and with the use of regularized regression, it is possible to detect robust changes over the course of 21 years but not on a regional scale. We also conclude that X-BASE does not contradict the trends of GLEAM and ERA5 Land in most cases.

We have added the following sentences to the manuscript:

Line 174: "Repeating the analysis for the 2001 to 2021 period with the inclusion of X-BASE does not lead to new insights. The short trend estimation period in combination with the large internal variability on regional scale does not allow for the detection of robust trends (Figure SI 8)."

Line 260: "We repeated the analysis shown from the 2001 to 2021 time span, allowing the inclusion of X-BASE (Figure SI 7). We also find a strongly detectable signal in all observational products over this time span, with X-BASE showing the largest ETx7d change over the time period."

3. Resampling to 2.5 degree resolution is, as the authors indicate, conservative, and is a reasonable way to perform the study. But it also means that the analyses are looking at ET averaged across very large areas that include diverse land cover types and vegetation conditions. While it would be difficult or impossible to examine sensitivities to sub-grid scale vegetation in CMIP6, it should be possible to consider land cover sensitivities in ERA5-Land and GLEAM. Are the trends, in the reanalysis/hybrid products, being driven by particular land cover types? This could inform our interpretation of trend results and is relevant to the flash drought framing. While flash droughts can occur in any land cover type, flash drought processes are expected to be quite distinct in, for example, forests vs. croplands. Stratification by land cover type might also be relevant to the authors' comparison of trends in extreme ET vs. trends in seasonal JJA ET.

We do agree that it would be very interesting to further dive into sub-grid scale effects. However, detection and attribution on smaller scale would become very difficult due to the high internal variability. We therefore rely on larger scale metric.

Formally attributing trends to regional drivers, including land use change, is thus outside the scope of this study, but would be extremely interesting for follow-up work.

4. I have mixed feelings about the jump from Europe to global in the results section. On the one hand, it's obviously useful to have hemispheric and global analyses as a contribution to the literature, and the use of IPCC regions and figure types will be helpful for future assessment reports. But the global analyses raise so many questions. All of the process-informed work presented in the manuscript is for Europe. Leaping to global, we see an interesting distribution in trends in Figure 3 and mixed results across regions and climate zones in Figures 3 and 4. It feels unsatisfying to present these results without explanation. Practically every part of the world shows some kind of interesting combination of results--what's going on in the Amazon? The Sahel? Australia?--and it's not obvious that the explanations provided for Europe apply in these other places. So . . . on the balance of it, my instinct would be to turn this into a longer paper or two separate papers. Either dig into the global results in earnest, or present this first, short paper as an introduction of the method with application to Europe and then write another (potentially very high impact paper) that goes global in earnest. I'm not going to insist on that, as there's nothing that I can see that's "wrong" in the global results. It would just be my recommendation.

To assess additional regions, also in response to other reviewer comments, we have included additional analysis to the supplement (Figures SI 5 and SI 6). They are analogous to figure 1 but for regions in eastern Brazil and the central US. We agree that investigating further regions would be interesting and have now included different cases, however, we also believe conducting a detailed analysis of each region would be inappropriate for the study and scope of the journal.

In the central US, we see that water limitation can be a crucial factor deciding whether ETx7d will be high or not. The two strongest events in this region both occurred at relatively high levels of soil moisture. The flash drought of 2012 manifests itself in one of the lowest ETx7d since 1950. Overall, there also seems to be less of a clear relationship with temperature, as the two strongest events are not anomalously warm. In Eastern Brazil, we find a strong relationship to temperature and less preconditioning on water availability. High net short wave radiation is a consistent feature across all regions and events considered here.

We find no reason to believe that the method fails outside of Europe and thus believe there is value in providing this data. However, we have over

We have changed the wording regarding the paragraph discussing these aspects (lines 175-182)

Reviewer #2 (Remarks to the Author):

Review of "Detecting Anthropogenically Induced Changes in Extreme and Seasonal Evapotranspiration Observations"

□ Summary and key results

Motivated by the contribution of high evapotranspiration events to flash droughts, this manuscript investigates anthropogenically induced trends in extreme and summer mean evapotranspiration (ET) on a global as well as regional scale. The study is based on both climate model outputs from CMIP6 (piControl/historical/SSP5-8.5), used to construct ET trend distributions in a world with and without global warming, and three observationally-informed model products (ERA5, GLEAM, X-BASE). A Ridge regression model is used to estimate the forced ET trends. Comparing the observational trends to the historical and piControl trend distributions reveals robust positive trends in both extreme and northern hemispheric summer ET which are very likely induced by anthropogenic activity. In addition, the study analyses ET trends for the 45 SREX regions. A number of regions in Eurasia and South America exhibit robust positive extreme and JJA ET trends. The authors explain positive trends as a mix of rising temperatures, leading to an increase in vapour pressure deficit, and an increase in incoming shortwave radiation, caused by aerosol reductions in recent decades.

General Assessment

With its focus on extreme events, the presented analysis makes a relevant, significant, and novel contribution to our understanding of how ET — a central component of Earth's hydrological cycle — evolves under the influence of global warming. Especially the use of a sophisticated regression technique designed to minimise the notoriously confounding influence of natural variability adds to the novelty and quality of the work. While the statistical method seems sound, and the results are compelling, I am less convinced by several aspects of the interpretation of the results, and especially the claimed link to flash droughts, for which I feel the manuscript lacks supporting evidence. In terms of style, the manuscript is written in appropriate language and generally good English. Nevertheless, I found it difficult to follow on my initial read-through, in parts because of occasional contradictory statements that impede the flow of the argument. The text would benefit from a more effective use of paragraphs (or even subsections) to structure ideas. In some places, essential information appears to be missing. I provide more details on these and other concerns/suggestions in the following major and minor comments. Where applicable, I reference the corresponding line numbers which can be found in the attached version of the manuscript. □

We thank the reviewer for their positive evaluation and in particular for critical, constructive and extremely careful comments regarding weaknesses in the argumentation and analysis. We have adjusted the manuscript in accordance with these comments, and we reply in detail below.

Major Comments

1. Assumption that ET_{x7d} occurs when the soil is saturated and PET is met □

The maximum value of 7-day accumulated ET at a given location in a given year does not automatically imply a situation in which $ET = PET$. There are presumably many places, especially in arid and semi-arid regions, where $ET < PET$ all year around. Also the typical soil moisture values of 0.35 in Fig. 1g seem to point at some moisture limitations. Whether the $ET = PET$ assumption indeed holds could perhaps be checked with a quantitative analysis. Alternatively, if the authors consider $ET = PET$ to be an important condition for their objectives (e.g. for removing water limitation as a confounding variable (line 60), or for testing if projected PET increases manifest in ET increases (line 63+64)), then an additional filter for $ET = PET$ (or approximately equal to) could be applied to the data. I am however not convinced of the importance of the $ET = PET$ assumption, given my next comment.

This is a fair point. We have moved away from the formulation that $ET = PET$ during an ET_{x7d} day event. Taking the assumption that PET can always be met during an extreme event is indeed not necessary for the analysis. We adjusted the text accordingly (line 62)

2. Link between extreme ET, PET, VPD and flash droughts

□ It would be useful to provide a definition of (flash) droughts in terms of the variable by which they are identified. If relatively dry soils are a typical drought characteristic, as would be consistent with the claimed role of high ET in driving flash droughts, then I would expect that particularly the moisture-limited regions around the world are prone to experience flash droughts. In contrast, a full soil moisture storage should reduce the drought risk. If this reasoning is sound, then locations and times where $ET = PET$ are precisely not the ones that are particularly

relevant for flash drought occurrences. Thus, before concluding which regions are allegedly more drought-prone due to a positive ETx7d trend, one might want to add an additional check of the regional soil moisture conditions. After all, a positive ETx7d trend could also originate from a positive soil moisture trend, e.g. due to increased precipitation. □ I also got generally a bit confused by the discussion of the relationship between ET, PET, and VPD (and their link to flash droughts). What is the exact role of VPD in contrast to PET? In lines 29 to 36, the authors state that many flash droughts are driven by high PET, and then assuming $ET = PET$ during ETx7d events links trends in ET to trends in PET. But the physical explanation for these trends in the Conclusions section only involves VPD. I think it would be helpful to clarify the various relationships between variables by presenting the equations that link them. □ Given the non-straightforward link between ETx7d and flash droughts (and the fact that no drought-specific analysis is performed), I want to point out that the investigation of anthropogenically-induced trends in ET variability and extremes is relevant for numerous atmospheric and land climate research questions, and therefore has great value by itself. It might be worth contemplating to assign flash droughts a less prominent role in the manuscript, e.g. by mentioning them as one phenomenon typically associated with ET extremes.

In response to this comment but also to another reviewer, we have removed a lot of the references to flash droughts, as they ended up being more confusing than helpful. The introduction now focuses more on why we hypothesize that extreme ET would be increasing. We link to research on precipitation, where this phenomenon is already more established and discuss increases in hydrological variability. Since these topics were the source of some confusion, we wanted to introduce them more formally.

We agree with the reviewer that our findings are relevant to a broader range of questions than just flash droughts. Flash droughts and extreme ET can be related but this is not always the case, as noted by the reviewer. The core concept linking this study to flash droughts is the fact that higher ET extremes allow for a faster transition from normal to dry conditions in those cases. This should directly lead to some faster onset flash droughts as the amount of ET can be the deciding factor in the onset.

We have focused the conclusions around this aspect.

Regarding the discussion of VPD, we have removed all references to VPD as we do not use the variable in the study. While relevant in the context of flash droughts and increasing ET, it is not a necessary concept to understand the results of the study. We now only refer to the increase in potential ET due to higher water holding capacity.

3. Presentation of 'observational data' and missing information in data section □

Except for X-BASE, none of the presented observational data provides information from direct ET measurements, and even in the case of X-BASE, spatial coverage is achieved by extrapolation using a ML model. (As such, it is also misleading that line 46 claims that we have accurate data since the late 20th century.) While I understand the use of these datasets as our 'best guess' of real ET, I think the text should be more transparent about their true nature, i.e. them being model products. Labelling them as 'observational' is fine as long as it is made clear that this is a simplification made for convenience. Beyond the transparency, I wonder if the limitations of these data sets have any implications for the results and conclusions. Perhaps the authors could comment on any relevant known biases. Lastly, I was missing some essential information about the observational dataset in the data section, namely: spatial and temporal coverage and resolution.

This is a fair comment that was also made by reviewer #1. We have added the following sentences to give more context (line 45):

"Observational products with global coverage can only be provided with the help of physical or statistical models. These models rely on a set of assumptions and contain known biases (Weerasinghe et al., 2020). Furthermore, these models often rely on additional data to augment ET data, adding additional uncertainty (Badgley et al., 2015). These aspects limit the amount of reliable observational data available for long-term analysis. These aspects limit the amount of reliable observational data available for long-term analysis, making the extraction of an anthropogenic signal increasingly challenging compared to variables such as temperature."

The referenced claim about accurate data was more regarding the advent of flux towers, and we have changed "accurate" to "more widespread". We have also added two sentences explaining some of the key challenges of observational ET products.

Additionally, we have repeated the analysis from 2001-2021 including X-BASE (Figure SI 7 and SI 8). The main conclusions made in this study are not put in question by this new analysis. We find a clear signal of change on the global scale, consistent with our findings so far, but no statistical significance from the regional analysis. This is expected due to the high amount of internal variability present when studying regional hydrological changes, given the short trend period for X-BASE. The manuscript now mentions these findings on lines 308-310.

4. Insufficient discussion of limitations inherent to methodology

□ The second paragraph of the Introduction dwells on how difficult it is to know anything about ET. Listing the

different problems at this point raises the expectation that the manuscript is going to present solutions to all these problems. However, the only actual 'solution' that it presents, is the use of a regression technique that tackles the confounding influence of natural variability. It would be more informative to limit the discussion of problems in the Introduction to those which the paper actually tackles, i.e. the extraction of the anthropogenic signals from natural variability and the use of ensembles and more than one observational dataset to limit the detrimental effect of model biases and other limitations. A more comprehensive discussion of remaining limitations could be moved to the data and methods or results section.

We state the research question which we aim to answer in the manuscript immediately following that section. We have added some of this discussion into the introduction, which now also encompasses a few important caveats about observational ET products used in the study (section in response to previous comment).

Further, we have added the following sentence to the discussion (line 379):

"The continuous effort to improve observational products is vital for studies such as this one, as the trends within the products used in this study are subject to substantial uncertainty. We attempted to navigate these uncertainties by checking whether the observational products agree with each other. Still, the trends within them should be interpreted with caution."

5. Generalisation of European context □

I was surprised to find such a strong focus on central Europe, given that Figure 1b in Christian et al. (2021) shows that the flash drought occurrence in Europe is rather low. Other regions with significant ETx7d trends, such as the northern part of South America, are known flash drought hot spots, rendering them potentially more interesting. This makes me wonder if the European focus was primarily chosen due to the authors' affiliation with European institutions. Should there be other reasons to only zoom in on Europe, they should be made explicit. Another issue with the Euro-centric view is that drivers of ET extremes might differ in different regions. The manuscript identifies heatwaves as the prevalent meteorological background of ETx7d. However, with their connection to stationary Rossby waves and suppressed convection, mid-latitude heatwaves may present themselves differently than tropical heatwaves. Similarly, the suggested role and timing of aerosol emissions seems specific to WCE. Aerosol emissions (or their reductions) might follow a different timing in other parts of the world which should be reflected in the respective ETx7d behaviour if the proposed mechanism is to be considered robust. Regarding the influence of aerosols, I stumbled over this paper by Zou & Yang et al. (2025) who find that aerosol reductions over China increase extreme precipitation which may have implications for ETx7d in the region. (<https://doi.org/10.1029/2024GL113887>) □ In summary: To which extent can the factors identified as 'drivers' of ETx7d be generalised? Can the regions showing significant anthropogenically-induced ETx7d trends be discussed in a joint framework?

The focus on Europe does primarily stems from the strong changes observed in this region, in addition to the affiliation with European institutions.

However, we have added analysis of additional regions and the following text to the manuscript (line 178).
"We find that strong positive anomalies in short-wave radiation are a crucial feature, also in other regions like eastern Brazil (Figure SI 5) and the central US (Figure SI 6). In dryer regions like the central US, we also see a preconditioning on water availability. The two strongest events occurred at relatively high soil moisture leading into the event."

In the Conclusion and Outlook section the mechanism behind the events is now described as follows:
"These events are primarily driven by warm spells with substantial atmospheric moisture demand and high incoming radiation but are contingent on sufficient water availability."

China is a very interesting case, with strong increases in earlier times and fast and significant decreases with the Clean Air Action Plan in aerosol optical depth over recent years. With the findings in Europe, we also expect increases in ETx7d in China driven by similar reasons as for extreme precipitation. The reason why it does not show up in the analysis is because ERA5 Land has a decreasing trend in this region where X-BASE and GLEAM both show significant and positive trends, in agreement with the CMIP6 climate models (see panel EAS in the figure below). We have no basis to not trust ERA5 Land in this specific case and making regional quality assessments of observational ET products is outside the scope of this study. However, it may be worth reconciling these trends with further data in future work, since we do expect a strong positive trend to emerge, like the climate models suggest.

For conciseness, we did not include this discussion in the manuscript.

Figure 1: Same as Figure 2 in the manuscript but for Eastern Asia (EAS), Southern Asia (SAS) and South East Asia (SEA). This figure was not added to the supplement.

6. Discussion of future scenario inconclusive and lacks quantitative analysis

I assume that the future SSP5-8.5 scenario was included to get a feeling for whether identified historical trends should be expected to continue into the future. However, given the strong model disagreement and regional dependence of future ETx7d, I am not sure what to take away from this discussion. Further, the claim that water limitations drive a decrease in ETx7d towards the end of the century (line 168 + 207) is not backed up by a quantitative analysis. Perhaps the discussion of future ETx7d could be dropped in favour of a more detailed explanation of the Ridge regression technique, which to me seems like a strong point of the work, as well as a more globally valid discussion of potential ETx7d trend drivers?

As mentioned by the reviewer, the question of whether this variable is projected to change substantially under a high emission scenario is an expected one. Even though the answer is not clear in this case, we believe it is worth to report it, rather than to discard it due to the absence of a clear trend. We also hypothesized that the climate models may agree better on the relative changes in ETx7d than in the mean as is the case for changes in mean and extreme precipitation. This is because the changes in the mean are constrained by more factors than changes in the extremes. For the extremes, increasing water holding capacity with higher temperature could become the most important feature.

It is fair that we do not provide quantitative evidence for the aspect of water limitation towards the end of the century. However, there is literature on future water limitation in CMIP6 models over Europe. Some models, like UKESM, show a very strong drying trend, especially in a high emission scenario. We have added citations to some of these studies.

Finally, we have added further explanation regarding Ridge to the methods section (Section 2.3). However, we also note this framework has been applied before in the studies cited, where it is also explained in more detail.

Minor Comments

Line 7 + 8: Given that ET was already identified as a flash drought driver in other studies, it is perhaps rather the trend in flash droughts that has not been looked at through the lens of ET?

This is a fair point. We have reworded the abstract to be consistent with this comment.

Line 9: what is the novelty regarding flash droughts? Did the study identify new regions that are at flash drought risk? Or does the lack of clear ETx7d trends in many of the flash drought prone regions around the world mean good news for these regions?

We now focus less on flash droughts in the introduction since the link between ETx7d and flash droughts is not entirely obvious. The discussion on flash droughts is now mostly limited to the discussion and outlook section.

Line 14: That the sign of ET trends depends on water supply and demand is a very generic statement. Apart from that, the evolution of supply and demand were not studied and compared to the trends.

This is a fair assessment. We have removed this part and instead added that the drivers vary regionally, which is also generic but more accurate.

Line 14+15: universal increase in ETx7d is not true, see the many white tiles in Fig. 4a as well as statements in line 262 + 263. Further, claim of water demand as the reason for the increases is just a claim without proof.

It is true that the current paper does not aim to demonstrate this. However, we base this statement on previous literature. We have added a bit more context, citing studies that find water to be limiting ET trends.

Line 29 + 30: Not sure about this presentation of cause and effect (although I admit that a discussion of circular mechanisms is always challenging). My attempt: Increasing temperature T increases the atmosphere's water holding capacity by raising the saturation vapour pressure q_s . Rising T and q_s together lead to an increase in PET. By how much VPD increases (or RH decreases) depends on both q_s and the response of ET to the altered atmospheric conditions (when the soil moisture is abundant, it will increase less than if the soil is moisture-limited). This is where equations for PET and VPD would come in handy.

The introduction is now more streamlined, with a focus on increases in potential ET due to increases in water holding capacity.

Line 31: Really about 50% while the other 50% is due to a lack of precipitation. Perhaps this should be noted.

We have updated this.

Line 39: Why does the levelling trend occur, is it relevant to the present study?

Both studies argue for water limiting ET further increases. The introduction now mentions this.

Line 41: What is their definition of extreme ET, do you use it, why not? (Perhaps something to pick up on in the methods section)

This study was published after we had done the bulk of the analysis. The author employs a peak-over-threshold framework, which is also very commonly used for extreme value analysis. We believe that our block-maxima approach is valid as well and possibly easier to communicate, as it does not involve the choice of a threshold. We have chosen to remain with the currently analysis setup.

Line 42 + 43: no study investigated 'trends in' seasonal and extreme ET globally? Given that this work also does not study seasonal and extreme ET in itself, this would make more sense.

Absolutely! Thanks for pointing this out. With the major changes to the introduction, this sentence was removed.

Line 65: detectable increases in magnitude/frequency?

Magnitude. We have added this.

Line 66: missing word + what does "amplified onset" mean?

We generally mean faster transition from normal to dry conditions, which we have added to the text.

Line 68: Only CMIP6 is mentioned here but the central part of the manuscript is that it looks at "observations" and contrasts them with CMIP6, the latter providing a benchmark

We wanted to put more emphasis the observational data, as we rely on this data more heavily in the study. We have expanded the section discussing climate models in the introduction (line 50):

"Secondly, ET is difficult to accurately model in climate simulations because it is highly sensitive to the representation of soil and plant dynamics in these models (Giardina et al., 2025). Climate models often diverge in their future projections of ET and soil moisture (Berg & Sheffield, 2018). Furthermore, ET is affected by the

stability of the atmosphere, which is only approximately represented in climate models.”

Line 71: What is the motivation to look at seasonal ET? How is it relevant to flash droughts? + should be ‘global and hemispheric’ not ‘global or hemispheric’, or just ‘global’

The idea is to see if they evolve differently, which is an idea stemming from studying precipitation changes. For precipitation, the extremes increase faster than the mean, which constitutes an increase in variance. We want to explore whether we can find this behavior in ET as well.

With the changed introduction we now discuss these aspects (line 25-30).

Line 75: short description of the different simulations missing (historical/piControl/SSP5-8.5, with time periods and temporal resolution). Temporal and spatial coverage and resolution are also lacking for the observational datasets.

We have added this to the Data section.

Line 79: why normalisation if study anyway about trends? Does the 1850-1950 period come from historical or piControl runs?

From historical. We have now clarified this in the text.

PiControl runs are generally not referred to by year as there is nothing that anchors them to a specific time period. Normalization helps the regression model, by removing baseline discrepancies in ET between climate models as well as climate models and observations. It can also be understood as fitting an intercept for every climate model individually. Since we’re not interested in studying these differences, we remove them.

We have added the following text:

“However, this has no effect on the trend within these datasets, which this analysis focuses on and is merely done to facilitate fitting the regression model.”

Line 81: climate model ‘simulations’

Changed.

Line 85: contradicts the claim that we need not worry about soil moisture because ETx7d represents cases where soils are fully saturated anyway (which I doubt, see major comments). Apart from that, I would expect that the focus on extremes, as well as the invoked dominance of atmospheric demand, makes land-atmosphere interactions particularly important.

We do not want to give the impression that water limitation can not be impactful for ETx7d. We have changed the text at various points to be clearer on this. This sentence now reads:

“Although these challenges complicate the identification of robust trends in ET observations, we hypothesize that the constraint of water limitation is weakened when focusing on high ET extremes, reducing the impact of the confounding variable.”

Line 91: X-BASE provides data ‘since’ (not ‘for’) 2001?

Yes, this was a typo that slipped through. We have changed it now to state the 2001 to 2021 period.

Line 94: why use a “wrong” compute baseline rather than no baseline for observational data?

Since we’re interested in the changes, and the regression model was trained on anomalies. The wrong base period has no impact on the resulting trends but makes the data easier to work with. The manuscript now mentions this.

Line 98: which variables? CMIP6 variables are not shown, however, this could be a nice additional analysis to show because it would corroborate the physical soundness of the simulations.

The variables shown in Figure 1. The mention of CMIP6 is a remanence of a previous version of the manuscript. This analysis was later cut, as it addressed a different research question. We have clarified the corresponding sentence.

Line 102: remote ‘and’

Line 103: effects ‘of’

Fixed.

Line 106: Is the ensemble mean ETx7d also a global map? At some point, spatial averaging is needed to arrive at a trend number (unless each grid cell contributes a trend value). When does this averaging happen? Perhaps it would be generally helpful to provide more information on how the algorithm goes from a map of ETx7d to a map of coefficients, and from there to a number. A schematic or equation might help.

The regression model's coefficients provide a spatial weighting that avoids regions of high variability. This is the step where the spatial averaging occurs. We have added some additional explanations to the manuscript. Such a schematic exists in de Vries et al., which is referenced in this section.

de Vries, I. E., Sippel, S., Pendergrass, A. G., and Knutti, R.: Robust global detection of forced changes in mean and extreme precipitation despite observational disagreement on the magnitude of change, *Earth Syst. Dynam.*, 14, 81–100, <https://doi.org/10.5194/esd-14-81-2023>, 2023.

Line 109: What does “ensure robustness across a variety of distributions” mean?

Every climate model in the CMIP ensemble represents its own distribution of values which the simulated variables can take. We can also consider every observational product an individual distribution. We want a statistical model which produces reliable results across all these different distributions. To gauge the extent to which a statistical model can generalize to a new distribution, we test it with an out of sample distribution. Or simply, it's guarding against overfitting.

Line 110: didn't understand, generally: naming consistency would help the reader, I found the following names that (I think) referred to the Ridge regression model throughout the text: model, statistical model, Ridge regression, regression model, linear model

We have added further clarification to a few mentions of “model” which were missing. As a result of this we have abandoned the terms Ridge, statistical, and linear model in favor of regression model to differentiate it clearly from climate models or other statistical models that are used for the generation of the observational ET products.

Line 113: General advice on avoiding “This”, especially at the beginning of sentences, because what it refers to is often ambiguous. Other instances of this problem in lines 118 (which procedure?), 160 (which period?), 201 (which events?), 348 (what?), 360 (what?), 361 (which phenomenon?)

I have updated the cases mentioned and read more clearly. We have also checked other cases and adjusted them when we felt like the interpretation could be unclear. In the cases where “this” refers to the most recent point mentioned in the last sentence and should not cause any confusion when considered in context.

Line 114: What does conservative model mean here? Does it provide a smaller range of trends than a less conservative version would?

Yes, we have added this.

Line 115: chosen or found? I didn't really understand what the regularisation is or does and this made me wonder if it is something to pick or whether it is determined by running some optimisation algorithm. Line 120-123: I would provide the information on why Ridge regression wasn't used again already here instead of in the results section.

Chosen. We have expanded the method section to include more information about the regression model (Section 2.3). We do not employ this method for regional trends, as one of the big benefits of this method (being able to identify regions with high signal-to-noise ratio) is less useful on regional scales, as the regions are often too small. It would technically be possible, but it has not yet been established, and we imagine there to be little insight generated. We therefore select a simpler methodology. This is now mentioned on line 265.

Line 124: 'both' the historical and ... □□Line 124: I would stick to the short hand piControl rather than spelling out pre-industrial for the same kind of consistency as always calling the regression model by the same name.

This should now be clearer.

Line 126: Rather than checking the physical understanding, Equation 2 checks for physical consistency. A small suggestion would be to swap the order of Equation 1 and 2 because then they would match the order of lines of evidence outlined here.

Done. Thanks for the suggestion.

Line 130 + 131: Sentence hard to understand, my alternative suggestion: "This is done by finding the quantile in the historical CMIP6 trend distribution to which the observational trend corresponds."

We have changed this sentence. Thanks for the suggestion.

Line 133: Equation 3

Line 152: Perhaps cite a heatwave review paper for this claim? E.g. Domeisen et al. (2023)

Added.

Line 160: Are aerosol effects and changes represented in all CMIP models? How do piControl and SSP5-8.5 handle aerosol emissions?

Yes, although their representation can be quite different and is a source of climate model uncertainty and disagreement. piControl has no changes in aerosol concentration. Aerosol emissions in SSP5-8.5 correspond to the level of economic activity and fossil fuels used with some assumption on future changes in terms of clean air regulation.

Rao, S., Klimont, Z., Smith, S. J., Van Dingenen, R., Dentener, F., Bouwman, L., Riahi, K., Amann, M., Bodirsky, B. L., van Vuuren, D. P., Aleluia Reis, L., Calvin, K., Drouet, L., Fricko, O., Fujimori, S., Gernaat, D., Havlik, P., Harmsen, M., Hasegawa, T., Heyes, C., Hilaire, J., Luderer, G., Masui, T., Stehfest, E., Strefler, J., van der Sluis, S., and Tavoni, M.: Future air pollution in the Shared Socio-economic Pathways, *Global Environ. Change*, 42, 346–358, <https://doi.org/10.1016/j.gloenvcha.2016.05.012>, 2017.

Line 164: I'm not convinced there is a clear increase in the observations for MED given the large variability of observational values

The increase since 2000 is something that all observations agree on. We make no statement about climate change being a central driver or whether this is part of a long term change. We added a sentence about high variability in this region as a caveat, as well as a citation of Vicente-Serrano et al.'s recent work.

Figure 1: Given that there is space, I would spell out all variable names in the subplot titles for consistency (had to guess what SAT and thought of saturation at first before realising that it must be surface air temperature..). Soil moisture = relative soil moisture saturation?

We have changed the labels accordingly. In remaking this figure we have found a slight coding error, only affecting figure 1. The main difference to the previous version is that ranking of the years changed and now 2015 is the second strongest year and 2022 is on rank 4. The interpretation remains the same. This also does not affect other figures, as the computations for figure 1 are not used for the further figures.

Line 168 + 194: Can this water limitation be seen in the data? E.g. in soil moisture just before ETx7d which declines towards later years and more so for models with stronger negative anomaly? Why is water limitation not/less visible in MED and NEU? (Same issue in Line 345)

It is quite established that some climate models, such as UKESM, have the tendency to become very warm and dry towards the end of the century. Other models, like MIROC have the tendency to be very energy limited, where ET can increase to possibly unrealistic degrees. Since these two models span the range of model projections, we are quite confident that water availability is one of the most important factors for projecting ETx7d into the future. Investigating soil moisture explicitly would be very laborious, as there are substantial differences in the way climate models represent it.

Water limitation is less visible for MED as there is likely less change in the water that is available for ETx7d. If this region is already semiarid with intermittent precipitation, we do not expect to see strong changes in the extreme (unless the precipitation regime changes really massively).

For NEU, we do not expect water limitation to become noticeable until the end of century, since this is a robustly energy limited region. Therefore, we expect ETx7d to increase due to the increased demand and this behavior is exactly what is visible in the figure.

Line 189 (caption Fig. 2): 'ETx7d' intensity 'anomaly'

We have removed the intensity to match the caption label in the figure.

Line 192: perhaps better for the reader to know the dates that day 125 and 225 correspond to, this gives a better feeling for the ETx7d season.

Good point!

Figure 2: the visibility of individual model spreads is in parts rather poor. I'm not sure what the best thing to do would be. If the journal allows for a bigger figure, one could try splitting each subplot in two and only show two to three models at a time. Alternatively, providing the model means as coloured lines might help to guide the eye? Or choosing different colours?

We have tried alternative visualizations, however, they turned out to be more confusing to the reader. Since we primarily aim to contrast models (in general) versus observations (from three sources), we think it is acceptable if the overlap between the individual models is not as clearly visible.

Line 199 + 200: An intensification is only really visible for GLEAM. X-BASE has a positive jump but preceded by a number of years with a negative trend, so not sure this is signal or noise. ERA5 Land line not visible at all.

ERA5 Land is indeed only barely visible. However, most of this is systematically covered in figure 4.

Line 201: What is the motivation to look at the timing? Was there an initial expectation? What is the reader supposed to take away from this analysis?

Overall, the idea of this portion of the analysis (Figure 1 and 2) is to give a basic understanding of what ETx7d events are, since these events are not commonly studied. In the beginning of this study, we hypothesized that water limitation would lead to a change in timing towards earlier in the season when more water is available, but this cannot be robustly concluded from the data that we analyzed. We have added this clarification to the manuscript.

Line 213: Shift in the distribution of ETx7d?

Line 226: 'This' (not 'That')

Fixed.

Line 227 - 229: "This results in" makes little sense in this context. Also just for clarification: does one end up with an ensemble of ensemble-mean estimates from the statistical model?

We have adjusted this section to read more clearly.

Yes, we get a prediction of the ensemble mean from every ensemble member individually.

Line 232: coefficients are not generally positive, there are some negative ones, so perhaps better 'mostly positive'?

This is a good point; we have changed this accordingly.

Figure 3 a - c: What is the meaning of the Beta values? □Also, interesting bimodality in the historical distributions. Any idea where this comes from? Has it perhaps any connection to the Wet-get-wetter-dry-gets-drier paradigm? □Caption: Line 217: obsolete colon, line 218: two c)

We have added two sentences that give some intuition about their meaning.

"The coefficients of the regression models, shown in Figures 3a, b, and c, are mostly positive. They can be mapped out to their corresponding location, showing where the model is drawing information from. Grid cells that are assigned a large coefficient contribute more to the prediction of the forced response. These grid cells are in regions where internal variability is comparatively low and where climate models agree best on their relationship with the ensemble mean."

The bimodality stems from the underlying distribution of climate models that have a varied response of ET to climate change. All climate models project an increase over the historical period, but the strength of the response varies. Where this bimodality comes from is somewhat unclear, but I would hypothesize that it is in the balancing of the turbulent heat fluxes. Some climate models dissipate the increased surface energy more via the sensible heat flux (smaller change in ET) and some over the latent heat flux (bigger change in ET)

No, I would not directly connect the pattern of the regression model to this paradigm.

Line 250: What follows after "average." was pretty difficult to grasp, had to read it several times. Perhaps this could be made more clear by not jumping back to Fig. 3 d + e but discussing this when first discussing these two subfigures figures?

This section points out the similarity in the shape of the distribution in Fig. 3d and e. The main difference stems from the fact that all ETx7d trends span a wider range. I have adjusted the section and hope that it reads more

clearly now.

Line 254: This would be natural place to start a new subsection

We have added subsections to the Results and Discussion.

Line 255: mean of ETx7d and JJA ET

Added.

Line 267: do not disagree = agree

Changed.

Line 268: worth naming these regions and perhaps contrasting them with Fig. 1 in Christian et al. (2021)? How does increase in ETx7d correlate with drought occurrence? Perhaps this manuscript says not only something about regions potentially at increased drought risk in the future but also something about already drought-prone regions for which a deterioration of the situation is not expected?

The regions prone to flash drought risk right now tend to already be more water limited now, which often do not see increases in ETx7d. However, just because in most years there is not enough water to produce a positive ETx7d trend does not mean that the overall increase could not lead to higher flash drought risk, as there could still be a faster transition from normal to dry.

For these cases it would likely be instructive to consider rarer extremes than a yearly block maximum.

Line 278: Is it possible to show PET directly to corroborate the claim that ET trends follow PET trends? And how does the claim that ET increases with PET fit together with the statement in the introduction that PET and ET have been shown to evolve differently in a warmer climate?

Thanks for pointing out this inconsistency. We have adjusted the wording. We have decided not to pursue quantifying PET for this study but it would be a very interesting metric to compare to in future work.

Line 287: indication = indication? + How can a trend in the extremes be driven by trends in the mean? Extreme values enter the computation of the mean value, not the other way around. If anything, a trend in the extremes can drive a trend in the mean. I suspect what is meant by this statement is that the entire distribution is simply shifted to higher values. However, the next sentence already contradicts such a notion because extremes increase more strongly than the mean. (Same issue in Line 342)

We did indeed mean a shift in the entire distribution, diagnosed by shift in the mean, which then increases the extreme values by the same amount. However, this also does not fully explain the changes that we see in the data. The change in extremes exceeds the change in in the mean but we hypothesize that is rather driven by an increase in hydrological variability, like what we see for precipitation. Therefore, it is a combination of shifting the mean and a widening of the distribution.

The section now reads as follows:

“This agreement could indicate that the trends in the extremes are mostly driven by the trend in the mean, where changes in the whole distribution led to higher values at the tail end. However, ETx7d trends are larger in most regions than the ones for JJA ET in the observations (Figure SI 3) and climate models (Figure SI 4), indicating an additional increase in variability in the distribution of ET.”

Line 289: ‘Figure’ SI 4

Fixed.

Line 293: This is a trivial statement. If changes are detected on a larger scale then it is impossible not to have changes on a smaller scale since the smaller scale changes integrate to give the larger scale changes. The statement only works the other way around (from small to large scale): Only if regional scales cancel out on could end up with a situation in which there are only regional and no hemispheric changes.

While it is true that large scale changes are made up of a combination of small scale ones, we must consider internal variability. The large scale metric will be less variable, as variability tends to cancel out over large enough regions, making trends in the large scale more easily detectable. This has been shown for several variables, where it is easier to detect a large scale trends than regional ones, even though large scale trends are of course the sum of regional trends (Lehner et al, 2020). Large scale trends emerge from internal variability sooner than regional ones, even if the trend is of similar magnitude because the range of possible trends induced by variability is bigger on the regional scale.

When looking at large scale metrics we can also take advantage of signal-to-noise optimization, where we give more weight to regions which have less internal variability and weigh the grid cells in a way so that internal variability cancels out. This does not work as well on the regional scale, as there are less grid cells to take advantage of.

Therefore, we could imagine a scenario where we can detect a change on a large scale, which is made up of regional trends which are not significant when taken in isolation.

Lehner, F., Deser, C., Maher, N., Marotzke, J., Fischer, E. M., Brunner, L., Knutti, R., and Hawkins, E.: Partitioning climate projection uncertainty with multiple large ensembles and CMIP5/6, *Earth Syst. Dynam.*, 11, 491–508, <https://doi.org/10.5194/esd-11-491-2020>, 2020

Line 294: One 70-year record per cell of the historical simulations?

Yes.

Figure 5 caption: First line is not a sentence, Line 338: regions, Line 339: two 'in' What is the bin size for records on the x-axis? Given the different dot sizes, the bin size seems to be different for different datasets. Why?

Fixed, thanks for spotting these.

Every dot is a 2.5° grid cell but not every region contains the same amount of 2.5° grid cells, which explains the differing number of points and their size.

Section 4: Summaries and conclusions are typically written in past tense, at least for the bits that recapitulate what was done in the study.

We have adjusted this.

Line 310: proof lacking for heatwaves being the primary driver in non-European cases

Changed the wording to be more accurate for other regions.

“These events are primarily driven by warm spells with substantial atmospheric moisture demand but are contingent on sufficient water availability. In addition, rising temperatures contribute to higher vapor pressure deficits, allowing for more rapid evaporation.”

Line 313: Would summarise approach again: “By comparison with the trend distributions constructed from forced and unforced CMIP6 ensemble simulations, we detect...”

Added.

Line 314: How were these likelihoods determined?

Based on the quantiles of the observed trend in the PiControl trend distribution and the p-values of the Mann-Kendall test, as shown by equations 5 and 6.

Line 350: factors

Fixed

Line 350 + 351: Surface warming is also not spatially homogeneous and VPD depends on the response of ET, horizontal moisture advection etc...

This is true yes, surface warming and VPD show some spatial variation, but they are universally positive in comparison to changes in ET or soil moisture, which can be positive or negative. While the changes in ET and soil moisture do affect changes in VPD to second order, the bulk of the change is given by unidirectional surface warming.

Line 353 - 356: Aerosols certainly play a role but there are many other factors that are likely important, too, such as soil moisture, precipitation changes, changes in wind and plant covers, ..

We added that this is only one aspect. Further aspects are discussed in the section following.

Line 360: Fig 2 does not show a simple return to 1950 levels of ET

Overall, we agree with the reviewer that Fig 2 does not suggest this. However, considering the evolution of ACCESS-CM2 or MPI-ESM1 in isolation does allow for this interpretation. We do not want to argue that this is the

case but simply caveat that we currently cannot disprove this interpretation. We have added the following sentence as clarification:

“However, when considering the magnitude of the recent observational changes, it becomes unlikely that they have been driven by changes in radiation only but most likely a combination of effects.”

Line 365: dynamical changes of what?

To better understand the effects of climate change, it is common to split up the signal into a thermodynamic and a dynamic component. The thermodynamic component relates to direct effects of increased temperature (such as increased water holding capacity of the air). The dynamic component relates to changes in the atmospheric circulation, such as shifting weather patterns. The study cited here finds that changing dynamics have contributed to an increase in heatwaves in Europe. In Europe, heatwaves are usually associated with high ET, therefore a dynamical change towards more heat waves would also increase (extreme) ET.

The sentence has been split up to include the following clarification.

“Furthermore, it is plausible that regional dynamically induced heat extreme trends, as in changes in circulation patterns, over Europe have contributed (Singh et al, 2023). Increases in heat waves would also increase VPD and exacerbate the trend of mean and extreme ET in this region.”

Line 372 + 373: Where were the effects of increased VPD visualised?

It is difficult to causally visualize these effects but since VPD is rapidly changing, we hypothesize that increases in ETx7d are mostly driven by increases in VPD. However, since we have no way of formally attributing this, we have changed the wording in the text.

Line 373 - 375: Are the regions highlighted in this manuscript consistent with the spatial changes in flash drought frequency presented in Zheng et al. (2023)?

We do find agreement in Northern South America, (South) Western Africa, and Central Europe. We do not find agreement over China and India, where they find strong increase in flash drought risk, and we find no changes. As discussed above, we do expect an increase of ETx7d over China to emerge in the near future, which would align with the finding of Zeng et al.

Line 376: what does “strengthened onset” mean? Earlier/more rapid/..?

More rapid. I have changed it to “speeding up the onset”.

Figures S13 and S14: It would be helpful if all axes would span the same ranges.

Done.

Reviewer #3 (Remarks to the Author):

This manuscript investigates long-running records of evapotranspiration to detect trends in extremes. The datasets include observation based records as well as climate model records. The modeled datasets further include a control dataset that is modeled for a world without human influence on the climate. The authors focus on the 7-day running total of ET, and specifically its maximum anomaly from that records long-term mean for a given region in each year in the dataset.

This selection reduces the number of confounding factors that may influence ET. The authors detect trends in the two longer records of observational data that would be very unlikely to occur without the human induced climate change. The study gives some additional insight in regional trends and contribution factors. I find the methodologies and conclusions well supported and appropriate for publication after only minor changes.

Some more discussion would be appreciated of the disagreement in day of maximum ET between observations and models. The disagreements are fairly large, in the order of 1-2 months for the MED region. Can the authors provide more insight in what is happening here? From the presented data it is difficult to judge if the large differences are a result of only small changes in event ranking, or result from important methodological gaps between models and observations. Perhaps plotting the timing of the second highest 7-day period (that doesn't overlap) in the same plot could test if there is more agreement in the top-two periods than there is in just the maximum?

Since we extract the seven days of maximum absolute ET, there is a strong seasonal component to the variable. Therefore, in drier regions, only a few weeks to months are responsible for ETx7d events. However, there are known discrepancies in the seasonality of precipitation in climate models. The Mediterranean is the driest of the three regions shown in the figure. Since the ETx7d events take place when there is at least some water available,

these differences in the precipitation (and thus soil moisture) seasonality emerge again in the timing of the ETx7d events.

We have added this discussion to the manuscript.

Further minor comments are included in the annotated manuscript.

We have addressed these. Thank you for the suggestions.

Introduction

Comment 1:

We have added a citation to this paper. Thanks for pointing it out.

Comment 2:

The sentence now reads “An increase in observational annual mean ET” to make it clear that it is not the number of observations.

Comment 3:

The sentence regarding this comment has been cut from the manuscript. However, we have added a citation to this interesting article later in the introduction.

Comment 4:

Added the missing word.

Data and Methods

Comment 1:

We have added further information about the different observational products, including the data ranges.

Comment 2:

Fixed, by adding the data ranges.

Results and Discussions

Comment 1:

Thank you!

Comment 2:

The last two sentences of this section are now in their own paragraph.

Comment 3:

We have fixed this typo.

Comment 4:

We have changed these sentences be more precise. They now read as follows.

“The observational products fall into similar quantiles of the historical and PiControl trend distribution in for both JJA ET and ETx7d. The ETx7d trends are larger, but so is the range in the PiControl distribution, which reduces the signal-to-noise ratio. The observational trends of JJA ET are generally smaller, but so are the likely ranges of the PiControl trends.”

Comment 5:

This sentence has been changed to be clearer.

“When comparing the two maps, it is worth noting that all regions which show an increase in ETx7d also show an increase in JJA ET, with the exception of the Mediterranean.”

Comment 6:

This section has also caused some confusion with other reviewers and has been changed to:

“For the regions where JJA ET decreased, we universally find no robust change in ETx7d. This is likely caused by long-term decreases in water availability, which are less visible when focusing on high ET extremes. However, a decrease in the mean and no change in the extreme suggest an increase in span of the distribution, as ETx7d moves further from the seasonal mean. A greater variance in precipitation and evapotranspiration is the core process driving drought risk (Allan, 2023). The divergence of mean and extreme ET can again be seen in Figures SI 3 and SI 4 where regions with decreasing JJA ET show less change in ETx7d.”

Figure captions

Comment Figure 4:

We have added a grey label to the legend.

Comment Figure 5:

We have changed the caption to be in line with the suggestion.

Comment Figure SI 3:

Yes, we have added this information to the caption.

“The dashed line has a slope of 1 and an intercept of 0, to give visual support for comparing the sizes of the ETx7d and the JJA ET trends.”